# Genomic and transcriptomic evidence for descent from *Plasmodium* and loss of blood schizogony in *Hepatocystis* parasites from naturally infected red colobus monkeys

Eerik Aunin[1], Ulrike Böhme[1], Theo Sanderson[2], Noah D. Simons[3], Tony L. Goldberg[4], Nelson Ting[3], Colin A. Chapman[5,6,7], Chris I. Newbold[1,8], Matthew Berriman[1], Adam J. Reid[1]*

1 Parasite Genomics, Wellcome Sanger Institute, Hinxton, Cambridge, United Kingdom, 2 Malaria Biochemistry Laboratory, The Francis Crick Institute, London, United Kingdom, 3 Department of Anthropology and Institute of Ecology and Evolution, University of Oregon, Eugene, Oregon, United States of America, 4 Department of Pathobiological Sciences, School of Veterinary Medicine, University of Wisconsin-Madison, Madison, Wisconsin, United States of America, 5 Department of Anthropology, Center for the Advanced Study of Human Paleobiology, The George Washington University, Washington DC, United States of America, 6 Shaanxi Key Laboratory for Animal Conservation, Northwest University, Xi'an, China, 7 School of Life Sciences, University of KwaZulu-Natal, Scottsville, Pietermaritzburg, South Africa, 8 Weatherall Institute of Molecular Medicine, University of Oxford, John Radcliffe Hospital, Oxford, United Kingdom

* ar11@sanger.ac.uk

**Data Availability Statement:** The Hepatocystis sp. assembly can be retrieved from the European Nucleotide Archive, under the study PRJEB32891

## Abstract

*Hepatocystis* is a genus of single-celled parasites infecting, amongst other hosts, monkeys, bats and squirrels. Although thought to have descended from malaria parasites (*Plasmodium* spp.), *Hepatocystis* spp. are thought not to undergo replication in the blood–the part of the *Plasmodium* life cycle which causes the symptoms of malaria. Furthermore, *Hepatocystis* is transmitted by biting midges, not mosquitoes. Comparative genomics of *Hepatocystis* and *Plasmodium* species therefore presents an opportunity to better understand some of the most important aspects of malaria parasite biology. We were able to generate a draft genome for *Hepatocystis* sp. using DNA sequencing reads from the blood of a naturally infected red colobus monkey. We provide robust phylogenetic support for *Hepatocystis* sp. as a sister group to *Plasmodium* parasites infecting rodents. We show transcriptomic support for a lack of replication in the blood and genomic support for a complete loss of a family of genes involved in red blood cell invasion. Our analyses highlight the rapid evolution of genes involved in parasite vector stages, revealing genes that may be critical for interactions between malaria parasites and mosquitoes.

## Author summary

*Hepatocystis* parasites are single-celled organisms, closely related to the *Plasmodium* species which cause malaria. But *Hepatocystis* are distinct–unlike *Plasmodium* they are thought not to replicate in the blood and cause little or no disease in their mammalian

and sample accession number ERS3649919. The individual accession numbers for the contigs are: CABPSV010000001-CABPSV010002439. Accession numbers for the apicoplast and the mitochondrion are LR699571-LR699572. Illumina HiSeq 4000 RNA-seq reads, containing a mix of Piliocolobus tephrosceles and Hepatocystis sp. sequences can be found in the European Nucleotide Archive under study accession PRJNA413051. Other data and code are available from our GitHub repository: https://github.com/adamjamesreid/hepatocystis-genome.

**Funding:** This work was funded by National Institutes of Health (NIH; https://www.nih.gov/), USA grant TW009237 as part of the joint NIH-NSF Ecology of Infectious Disease program and the UK Economic and Social Research Council (TLG, NT, CAC). National Science Foundation Grant BCS-1540459 (NT, NDS; https://www.nsf.gov/). The Wellcome Sanger Institute is funded by the Wellcome Trust (grant 206194/Z/17/Z; https://wellcome.ac.uk/) which supports EA, UB, MB, and AJR. AJR is also supported by funding from the UK Medical Research Council (MRC Programme grant #: MR/M003906/1; https://mrc.ukri.org/). TS is supported by a Wellcome Trust Sir Henry Wellcome Fellowship (210918/Z/18/Z). CIN is funded by a Wellcome Investigator Award (104792/Z/14/Z; https://wellcome.ac.uk/). The funders had no role in study design, data collection and analysis, decision to publish, or preparation of the manuscript.

**Competing interests:** The authors have declared that no competing interests exist.

hosts. They are transmitted from one host to the next, not by mosquitoes, but by biting midges. In this study we generated a genome sequence for *Hepatocystis*–the first time this data has ever been produced and analysed for this species. We compared genome sequences of *Hepatocystis* and *Plasmodium*, confirming that *Hepatocystis* is descended from *Plasmodium*. We strengthened support for the absence of replication in the blood and, in line with this finding, discovered that genes involved in interaction with red blood cells have been lost in *Hepatocystis*. Our analyses revealed rapid evolution of genes which are active when the parasite is in the insect vector, highlighting those which might be important for understanding interaction between malaria parasites and mosquitoes. *Hepatocystis* has a fascinating evolutionary story and is a powerful comparator for understanding malaria parasite biology.

## Introduction

Species of the genus *Hepatocystis* are single-celled eukaryotic parasites infecting, amongst other hosts, Old World monkeys, fruit bats and squirrels [1]. Phylogenetically, they are thought to reside within a clade containing *Plasmodium* species, including the parasites causing malaria in humans [2]. They were originally considered distinct from *Plasmodium* and have remained in a different genus because they lack the defining feature of asexual development in the blood, known as erythrocytic schizogony [3]. The presence of macroscopic exo-erythrocytic schizonts (merocysts) in the liver of the vertebrate host is the most prominent feature of *Hepatocystis* [3]. Similar to *Plasmodium* parasites, *Hepatocystis* merocysts yield many single-celled merozoites. However, unlike *Plasmodium*, *Hepatocystis* merocysts appear to be the only replication phase in the vertebrate host [4]. First generation merozoites of *Plasmodium* spp. are released from liver cells and invade red blood cells, where they multiply asexually, before erupting from red cells as secondary merozoites. These merozoites invade further red blood cells before some develop into stages that can be transmitted to the vector. In contrast, liver merozoites of *Hepatocystis* spp. are thought to commit to the development of gametocyte transmission stages directly upon invading red blood cells. They are then vectored not by mosquitoes, but by biting midges of the genus *Culicoides* [5]. After fertilisation, *Hepatocystis* ookinetes encyst in the head and thorax of the midge between muscle fibres, whereas *Plasmodium* ookinetes encyst in the midgut wall of mosquitoes. After maturation, oocysts of both *Plasmodium* and *Hepatocystis* rupture and release sporozoites that migrate to the salivary glands [1]. These discrete biological differences, in the face of phylogenetic similarity and many shared biological features, make *Hepatocystis* a potentially powerful comparator for understanding important aspects of malaria parasite biology, such as transmission and host specificity.

A population of red colobus monkeys (*Piliocolobus tephrosceles*), from Kibale National Park, Uganda were previously shown to host *Hepatocystis* parasites based on morphological identification of infected red blood cells and DNA sequencing of the *cytochrome b* gene [6]. In this work we use *Hepatocystis* sp. genome and transcriptome sequences derived from *P. tephrosceles* whole blood samples to generate a draft genome sequence and gain insights into *Hepatocystis* evolution. We go on to use these insights to explore key aspects of malaria parasite biology such as red blood cell invasion, gametocytogenesis and parasite-vector interactions.

## Results

### Genome assembly and annotation

While examining a published genomic sequence generated from the whole blood of a red colobus monkey (*Piliocolobus tephrosceles*) in Kibale National Park, Uganda (NCBI assembly ASM277652v1), we noticed the presence of sequences with significant similarity to *Plasmodium* spp. We hypothesised that this represented genomic material from a bloodborne parasite captured during the sequencing process and for each contig in the assembly we examined the AT-content and sequence similarity to *Plasmodium* spp. and macaque (Fig 1A). We identified a substantial subset of contigs that appeared to be derived from an apicomplexan parasite. Phylogenetic analysis, using an orthologue of *cytochrome b* from these contigs, suggested that they represent the first substantial genomic sequence from the genus *Hepatocystis* (Fig 1B), a parasite previously reported in Kibale National Park that infects at least four species of Old World monkeys [6]. At least four species of *Hepatocystis* are known to infect African monkeys–*H. kochi, H. simiae, H. bouillezi* and *H. cercopitheci* [6] –but with little sequence data currently linked to morphological identification, it was not possible to determine the species. We have thus classified the parasite as *Hepatocystis* sp. ex *Piliocolobus tephrosceles* (hereafter *Hepatocystis* sp.; NCBI Taxonomy ID: 2600580). The extraction of the *Hepatocystis* sp. sequences from the *P. tephrosceles* assembly yielded a set of 11,877 scaffolds with a total size of 26.26 Mb and an N50 of 2.4 kb. Automated genome annotation with Companion [7] identified 2,967 genes and 1,432 pseudogenes in these scaffolds. To improve upon this assembly, we isolated putative *Hepatocystis* sp. reads from the original short read DNA sequencing data. These were assembled into a draft quality nuclear genome assembly of 19.95 Mb, comprising 2,439 contigs with an N50 of 18.3 kb (Table 1). The GC content (22.05%) was identical to that of *Plasmodium* spp. infecting rodents, and slightly higher than *P. falciparum* (19.34%). We identified 5,341 genes, compared to 5,441 in *P. falciparum* and 5,049 in *P. berghei*, suggesting a largely complete gene set (Table 1; S1 Table). Despite the fragmented nature of the assembly, we were able to identify synteny with *P. falciparum* around centromeres (S1 Fig) and evidence of clustering of contingency gene families (S2 Fig), as seen in most *Plasmodium* species.

### Phylogenetic position of *Hepatocystis* sp. ex. *Piliocolobus tephrosceles*

There is consensus that *Hepatocystis* spp. are nested within the *Plasmodium* genus [2,8,9], however their placement within the genus has not been robustly determined. Indeed, our *cytochrome b* phylogeny confirms that our assembled genome is that of a *Hepatocystis* species, but it provides little support for the placement of this genus in relation to *Plasmodium* spp. A phylogeny generated using all mitochondrially encoded protein sequences also provided little support for key nodes (S3 Fig; see GitHub page for all sequence alignment data). The mitochondrial genome is therefore not reliable for determining the species phylogeny. A phylogeny based on 18 apicoplast proteins was more robust and placed *Hepatocystis* sp. as an outgroup to the *Plasmodium* species infecting rodents (*Vinckeia;* S4 Fig). We wanted to reliably place *Hepatocystis* sp. relative to other *Hepatocystis* species. Limited sequence data are available for *Hepatocystis* outside of this study. However, 11 nuclear genes have been sequenced for *H. epomophori*, a parasite of bats [2]. Based on the sequence of these genes, we found that *Hepatocystis* sp. forms a sister group to *H. epomophori* (S5 Fig). Furthermore, the *Hepatocystis* genus again forms a sister group to the *Vinckeia* subgenus of *Plasmodium*, although the tree contains some ambiguous branch points. To improve the robustness of the placement of *Hepatocystis* sp. within *Plasmodium*, we used 2673 orthologous nuclear genes from each of 12 species across the *Plasmodium* genus, which robustly places *Hepatocystis* sp. as a sister clade to the

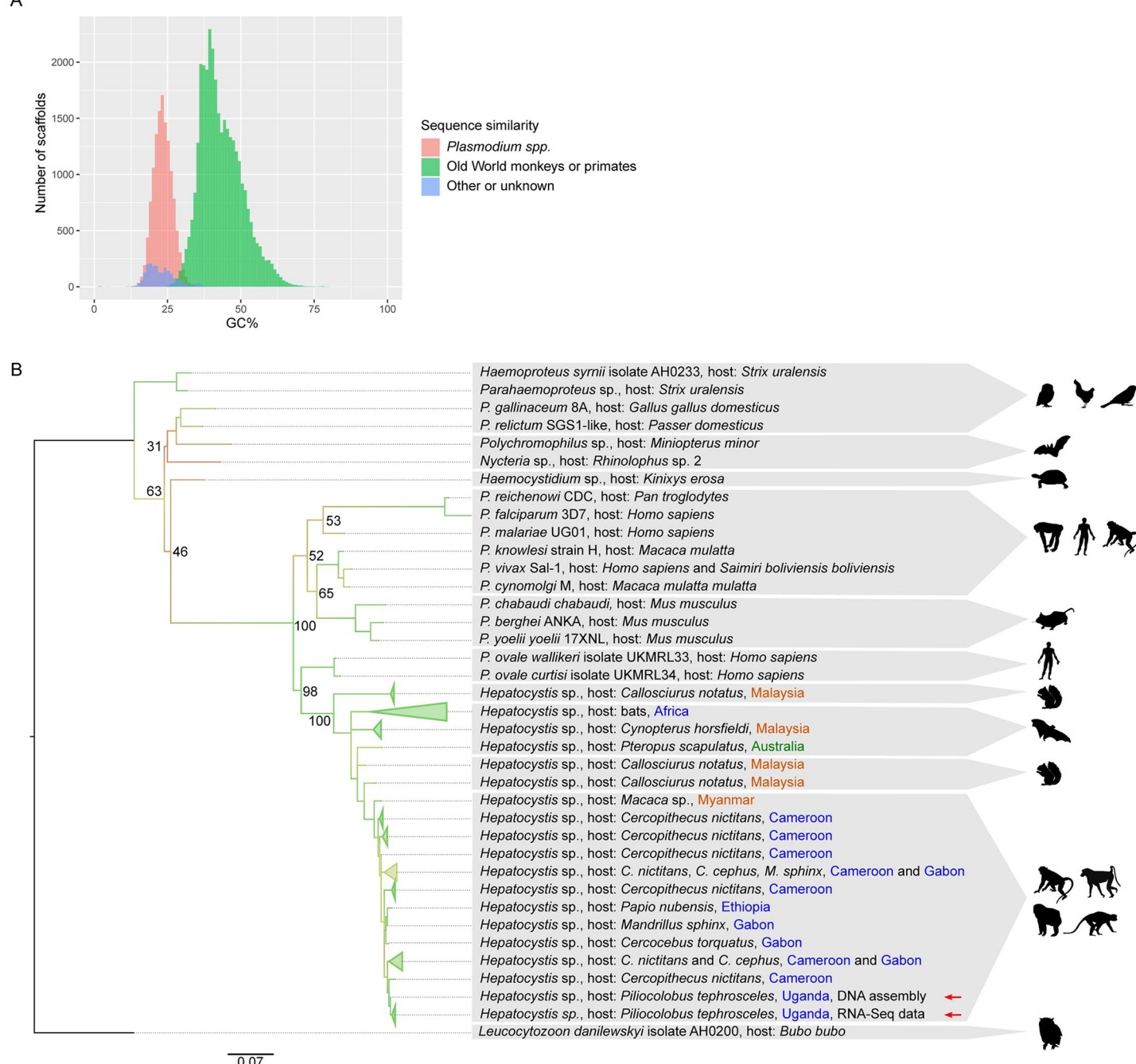

**Fig 1. An assembly of genomic sequencing reads from a red colobus blood sample contained significant amounts of sequence from the parasite *Hepatocystis* spp.**
(A) Contigs from the red colobus (*Piliocolobus tephrosceles)* assembly had a bimodal distribution of AT-content and sequence similarity to *Plasmodium* spp. (B). A phylogenetic tree of *cytochrome b* indicated that the closest match for the apicomplexan parasite sequenced from red colobus blood is a *Hepatocystis* isolate from a monkey host. Parasite *cytochrome b* sequences derived from RNA-seq assemblies from red colobus blood samples are almost entirely identical to the *cytochrome b* sequence assembled from *Hepatocystis* DNA reads from a single monkey. Branches of the tree have been coloured by bootstrap support values from 15 (red) to 100 (green). Some bootstrap support values are also shown next to the nodes as text. Red arrows highlight the *Hepatocystis* samples from the current study. Blue place names indicate the African continent, green Australia, orange Asia.

*Plasmodium* species infecting rodents (subgenus *Vinckeia;* Fig 2). Interestingly, some *Vinckeia* species (*P. cyclopsi*) also infect bats, supporting an earlier suggestion that the ancestor of

**Table 1. Features of the *Hepatocystis* sp. ex. *Piliocolobus tephrosceles* assembly compared to *P. falciparum* 3D7, *P. vivax* P01 and *P. berghei* ANKA.**

| | *Hepatocystis sp.* | *P. falciparum* 3D7 | *P. vivax* P01 | *P. berghei* ANKA |
|---|---|---|---|---|
| *Nuclear genome* | | | | |
| Genome size (Mb) | 19.95 | 23.3 | 29.0 | 18.7 |
| G+C content (%) | 22.05 | 19.34 | 39.8 | 22.05 |
| Gaps within scaffolds | 979 | 0 | 431 | 0 |
| No. of scaffolds | 2439 | 14 | 240 | 19 |
| No. of chromosomes | ND | 14 | 14 | 14 |
| No. of genes* | 5,341 | 5,441 | 6,650 | 5,049 |
| No. of pseudogenes | 28 | 158 | 158 | 129 |
| No. of partial genes | 1,475 | 0 | 196 | 8 |
| No. of ncRNA | 19 | 103 | 35 | 47 |
| No. of tRNAs | 41 | 45 | 45 | 45 |
| No. of telomeres | 0** | 26 | 1 | 12 |
| No. of centromeres | 5 | 13 | 14 | 14 |
| *Mitochondrial genome* | | | | |
| Genome size (bp) | 6,595 | 5,967 | 5,989 | 5,957 |
| G+C content (%) | 30.99 | 31.6 | 30.5 | 30.9 |
| No. of genes | 3 | 3 | 3 | 3 |
| *Apicoplast genome* | | | | |
| Genome size (kb) | 27.0 | 34.3 | 29.6 | 34.3 |
| G+C content (%) | 13.29 | 14.22 | 13.3 | 15.1 |
| No. of genes | 28 | 30 | 30 | 30 |
| *Completeness* | | | | |
| CEGMA—complete | 63.31% | 69.35% | 68.15% | 70.16% |
| as least partial | 69.35% | 71.77% | 71.77% | 73.39% |

* including pseudogenes, duplications and partial genes, excluding non-coding RNA genes

** two small contigs have telomeric repeats (scaffold_2410–5 telomeric repeats, scaffold_2364–9 telomeric repeats)

*Hepatocystis* and *Vinckeia* might have infected bats [10]. However, whole-genome data also suggest that *Vinckeia* is derived from a group of monkey-infecting parasites [11]. *Hepatocystis* sp. has a long branch that could indicate rapid evolution after splitting from its ancestor with *Vinckeia*.

## *In vivo* transcriptome data supports a lack of erythrocytic schizogony

Transcriptome sequencing of blood samples from 29 individuals was performed as part of the red colobus monkey genome sequencing project [12]. We found evidence that each of these individuals was infected with the same species of *Hepatocystis* as found in the genomic reads, consistent with high prevalence of this parasite in Kibale red colobus monkeys [6]. The extremely low SNP density suggested that parasites from different red colobus individuals were highly related (Fig 3A). We identified an average of 1.36 SNPs (standard deviation = 0.47) and 0.43 indels (standard deviation 0.15) per 10 kb of genome when calling variants using RNA-seq reads.

Although it is believed that *Hepatocystis* spp. do not undergo erythrocytic schizogony [3], this has been challenged by limited microscopic evidence for asexual stages in the blood [13]. To determine whether there was transcriptomic evidence for schizonts in the blood we

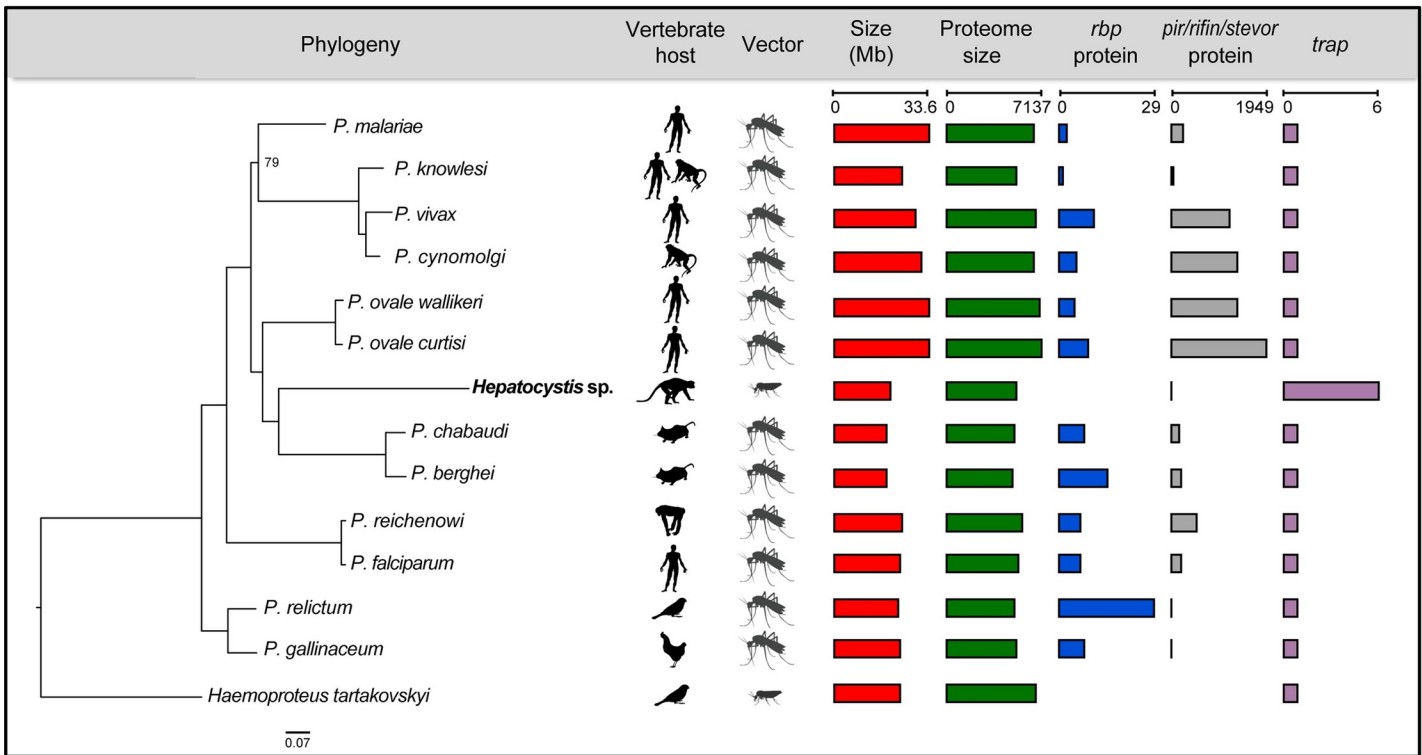

**Fig 2. Whole genome phylogeny and key features of the *Hepatocystis* genome.** A whole-genome phylogenetic tree is combined with a graphical overview of key features of *Hepatocystis* and *Plasmodium* species (genome versions from August 2019). The maximum likelihood phylogenetic tree of *Hepatocystis*, *Plasmodium* and *Haemoproteus* species is based on an amino acid alignment of 2673 single copy orthologs encoded by the nuclear genome. Bootstrap support values of all nodes were 100, except for one node where the value was 79. The rooting of the tree at *Hae. tartakovskyi* is based on previously published *Plasmodium* phylogenetic trees [11,45]. TRAP—thrombospondin-related anonymous protein. RBP protein—reticulocyte binding protein.

deconvoluted the *Hepatocystis* sp. transcriptomes using transcriptome profiles representing different *Plasmodium* life stages. We found no such evidence, but observed varying proportions of cells identified as early blood stages (rings/trophozoites) and mature gametocytes (Fig 3B; S6 Fig; S2 Table). It is thought that chronic infections (of up to 15 months) may be maintained from continual development in the liver [3]. The presence of early blood stages in these individuals may therefore reflect this continual production of new blood forms, rather than recent infection. Proportions of rings and trophozoites were positively correlated and both these forms correlated negatively with female gametocytes (Fig 3C). Interestingly, the inferred proportions of male and female gametocytes were not strongly correlated suggesting there might be variation in commitment rates of gametocytes to male or female development.

## Expanded and novel gene families

The largest gene family in *Hepatocystis* sp. was a novel family, which we have named *Hepatocystis*-specific family 1 (*hep1*; Table 2). These 12 single-exon genes (plus four pseudogenes) each encode proteins of ~250 amino acids, beginning with a predicted signal peptide (S7A Fig). We could find no significant sequence similarity to genes from any other sequenced genome (using HHblits [14]). However, they contain a repeat region with striking similarity to that in *Plasmodium kahrp*, a gene involved in presenting proteins on the red blood cell surface [15]. Three members were highly expressed in *in vivo* blood stages, with one correlating well with the presence of early stages (S8A Fig; HEP_00211100, Pearson's r = 0.80 with rings).

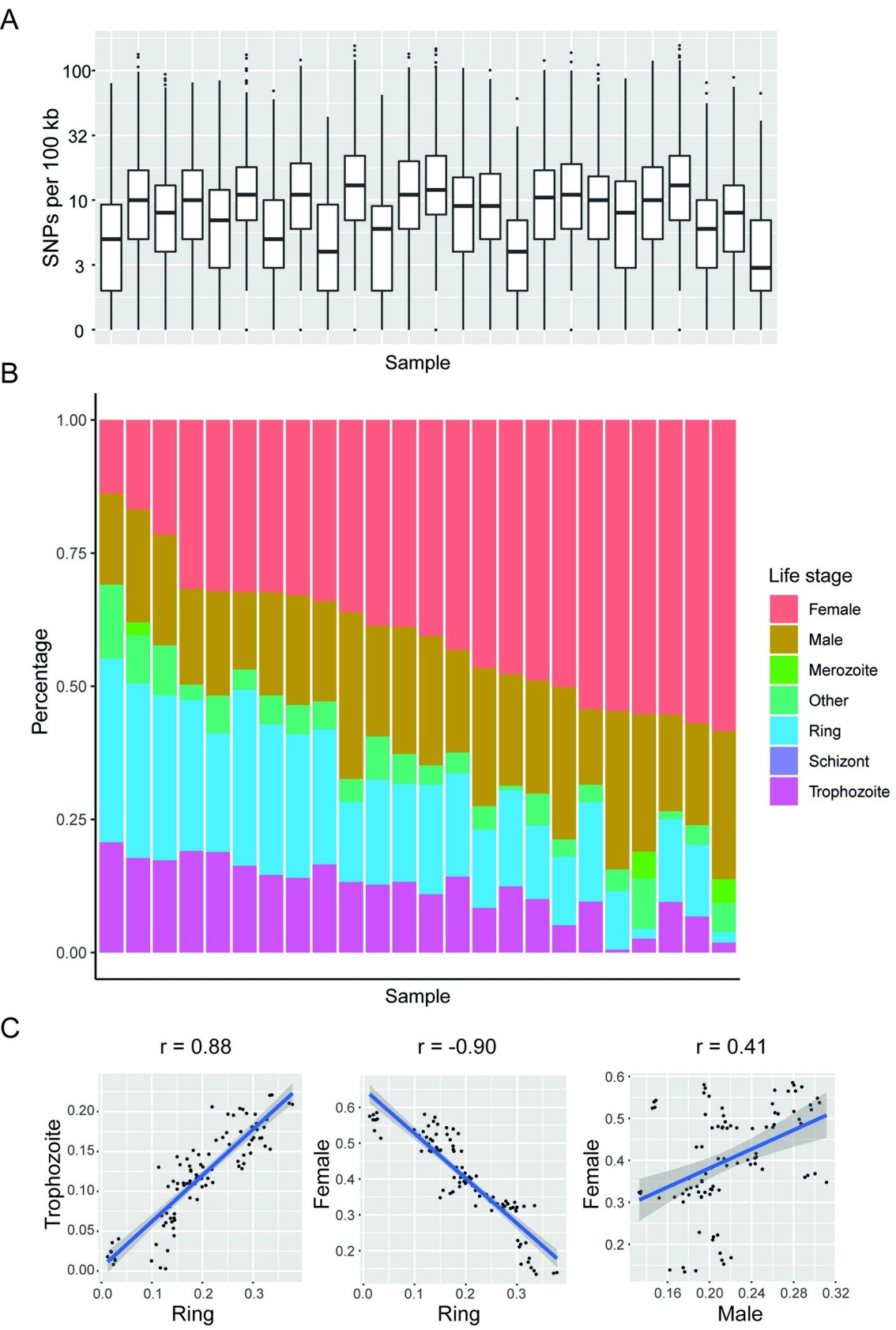

**Fig 3. *Hepatocystis* sp. *in vivo* RNA-seq data supports a lack of erythrocytic schizogony and a variable sex ratio.** (A) Distributions of SNPs per 100 kb in each *Hepatocystis* sp. RNA-seq sample, relative to the genome assembly reference, highlight consistently low genetic diversity. Samples SAMN07757853, SAMN07757863, SAMN07757870 and SAMN07757873 have been excluded from the figure due to their low expression of *Hepatocystis* genes. (B) Deconvolution of RNA-seq samples to identify parasite stage composition shows no evidence for blood schizonts. Ring and trophozoite cells are assumed to relate to early stages of gametocyte development, which are not distinguishable from asexual rings and trophozoites. (C) Proportions of early blood stages (ring and trophozoite) are negatively correlated with mature female gametocytes, however male and female gametocyte ratios are poorly correlated, suggesting that sex ratios vary among samples.

Another novel family, *hep2*, contained N-terminal PEXEL motifs, suggesting it is exported (S7B Fig). Distinct parts of its sequence showed similarity (albeit with low significance) to exported proteins from *P. malariae* (PmUG01_00051800; probability: 77.88% HHblits) and *P. ovale curtisi* (PocGH01_00025800; 78.47%) as well as a gene in *P. gallinaceum* (PGAL8A_00461100; 69.52%). Three or four members were highly expressed in blood stages *in vivo*, with one member highly correlated with predicted proportions of early blood stages (S8B Fig; HEP_00165500, Pearson's r = 0.83 with rings; HEP_00480100, Pearson's r = 0.77 with rings).

The gene encoding Thrombospondin-Related Anonymous Protein (*trap*) is involved in infection of salivary glands and liver cells by *Plasmodium* sporozoites. It is found strictly in a single copy in all *Plasmodium* species sequenced to date. However, it is present in six copies in *Hepatocystis* sp., suggesting that *trap*-mediated aspects of sporozoite-host interactions may be

**Table 2. Size of known and novel gene families in *Hepatocystis* sp. in comparison to its relatives.**

| | *Hepatocystis* sp. | *Plasmodium falciparum* 3D7 | *P. berghei* ANKA | *P. vivax* P01 | *Haemoproteus tartakovskyi* |
|---|---|---|---|---|---|
| ApiAP2 transcription factors | 27 | 27 | 26 | 27 | 20 |
| *Hepatocystis*-specific family 1 (*hep1*) | 16 (includes 4 pseudogenes) | 0 | 0 | 0 | 0 |
| *Hepatocystis*-specifc family 2 (*hep2*) | 10 (includes 4 pseudogenes) | 0 | 0 | 0 | 0 |
| *cpw-wpc* | 8 | 9 | 9 | 9 | 8 |
| 6-cysteine proteins | 7 | 14 | 13 | 15 (includes 1 pseudogene) | 22 |
| *lccl* | 6 | 6 | 6 | 6 | 6 |
| Thrombospondin-Related Anonymous Protein (*trap*) | 6 | 1 | 1 | 1 | 1 |
| *pir* | 5 (includes 1 pseudogene) | 0 | 218 (includes 83 pseudogenes) | 1212 (includes 109 pseudogenes) | 0 |
| Serine repeat antigen (SERA) | 5 (1 pseudogene) | 9 | 5 | 13 | 2 |
| early transcribed membrane protein (*etramp*) | 4 | 15 | 7 | 10 | 1 |
| exported protein 1 (*exp1*) | 4 (includes 2 pseudogenes) | 1 | 1 | 1 | 0 |
| Tryptophan-rich antigen | 4 | 4 (includes 1 pseudogene) | 11 (includes 4 pseudogenes) | 40 | 0 |
| Lysophospholipase | 2–4 | 6 (includes 1 pseudogene) | 5 (includes 2 pseudogenes) | 7 (includes 1 pseudogene) | 1 |
| Phist domain-containing | 2 | 81 (includes 19 pseudogenes) | 3 | 82 (includes 6 pseudogenes) | 3 |
| *fam-a* | 1 | 1 | 74 (includes 26 pseudogenes) | 1 | 1 (partial) |
| Reticulocyte binding protein (Rh/rbp) | 0 | 7 | 15 | 10 | 0 |

more complex. None of these genes were highly expressed in blood stage transcriptomes, consistent with their known role in sporozoites. Exported protein 1 (e.g. PF3D7_1121600) is a single copy gene in all *Plasmodium* species. It encodes a Parasitophorous Vacuolar Membrane (PVM) protein and is important for host-parasite interactions in the liver [16]. In *Hepatocystis* sp. it is expanded to four copies.

## Missing orthologues tend to be involved in erythrocytic schizogony

The genomes of *Plasmodium* spp. each contain large families of genes known or thought to be involved in host parasite interactions [17]. These include, amongst others the *var*, *rif* and *stevor* genes in the *Laverania* subgenus, *SICAvar* genes in *P. knowlesi* and the *pir* genes across the genus. We find only four intact *pir* genes and a single *pir* pseudogene in *Hepatocystis* sp., compared to ~200–1000 in *Plasmodium* spp. infecting rodents. One was particularly highly expressed in the blood stages from most monkeys that were sampled (S8C Fig; HEP_00069900). All of these are most similar to the ancestral *pir* subfamily, present in single copy in *Vinckeia* species and in 19 copies in *P. vivax* P01 (Table 2; S9 Fig). The best described role for *pir* genes is in *Vinckeia* parasites, where they are involved in establishing chronic infections in mice [18]. Given that asexual *Hepatocystis* parasites are not thought to exist in the blood of monkeys or bats, there would be no need for this function of *pir* genes. However, in *Vinckeia*, *pir* genes are expressed in several other stages, including male gametocytes [19], which do feature in the *Hepatocystis* lifecycle. The function of the ancestral *pir* gene subfamily is unknown, although it is expressed at multiple stages of the lifecycle in *P. berghei* [20].

We wanted to determine, more generally, the types of genes that might have been lost in *Hepatocystis* sp. relative to *Plasmodium* spp. This is made difficult due to uncertainty in determining missingness in a draft genome. We overcame this problem using clusters of genes identified as having common expression patterns across the lifecycle of the close *Hepatocystis* relative *P. berghei* [20]. Cluster 10 (late schizont expression; Fisher's Exact test odds ratio = 0.09, FDR = 0.0002) tended to contain orthologues shared by *P. berghei*, *P. ovale wallikeri* and *P. vivax*, but not *Hepatocystis* sp. (Fig 4; S3 Table; S10 Fig). Eleven out of 25 orthologues missing in the late schizont cluster (including two pseudogenes) encoded Reticulocyte Binding Proteins (RBPs). In fact, we could not identify any RBPs in the *Hepatocystis* sp. genome or *Hepatocystis* sp. RNA-seq assemblies. Interestingly, counter to previous suggestions [21], we found no evidence for RBP sequences in the *Hae. tartakovskyi* genome sequence. However, we did find fragmentary sequences with significant sequence similarity to RBPs from malaria parasites with avian hosts in transcriptome assemblies of *Leucocytozoon buteonis* [22] and *Hae. columbae* [23] (S11 Fig). Also missing from this cluster was *Cdpk5*, a kinase that regulates parasite egress from red cells [24]. The other principal gene families involved in erythrocyte binding and invasion by *Plasmodium* are the erythrocyte binding ligands (*eh*)/ duffy-binding protein (*dbp*) and the merozoite surface protein (*msp*) families [25]. These are largely conserved relative to *P. berghei*. Thus, orthologues missing relative to *Plasmodium* spp. tend to be involved in erythrocytic schizogony, the part of the life cycle also absent.

## The most rapidly evolving genes are often involved in vector biology and control of gene expression

Our whole-genome phylogeny (Fig 2) showed a long *Hepatocystis* sp. branch, suggesting that some genes have changed extensively in *Hepatocystis* compared to *Plasmodium* spp. This might indicate functional changes important for the particular biology of *Hepatocystis*. We found previously that the ratio of synonymous mutations to synonymous sites (dS) saturates between *Plasmodium* clades [45] and therefore considered the ratio of non-synonymous

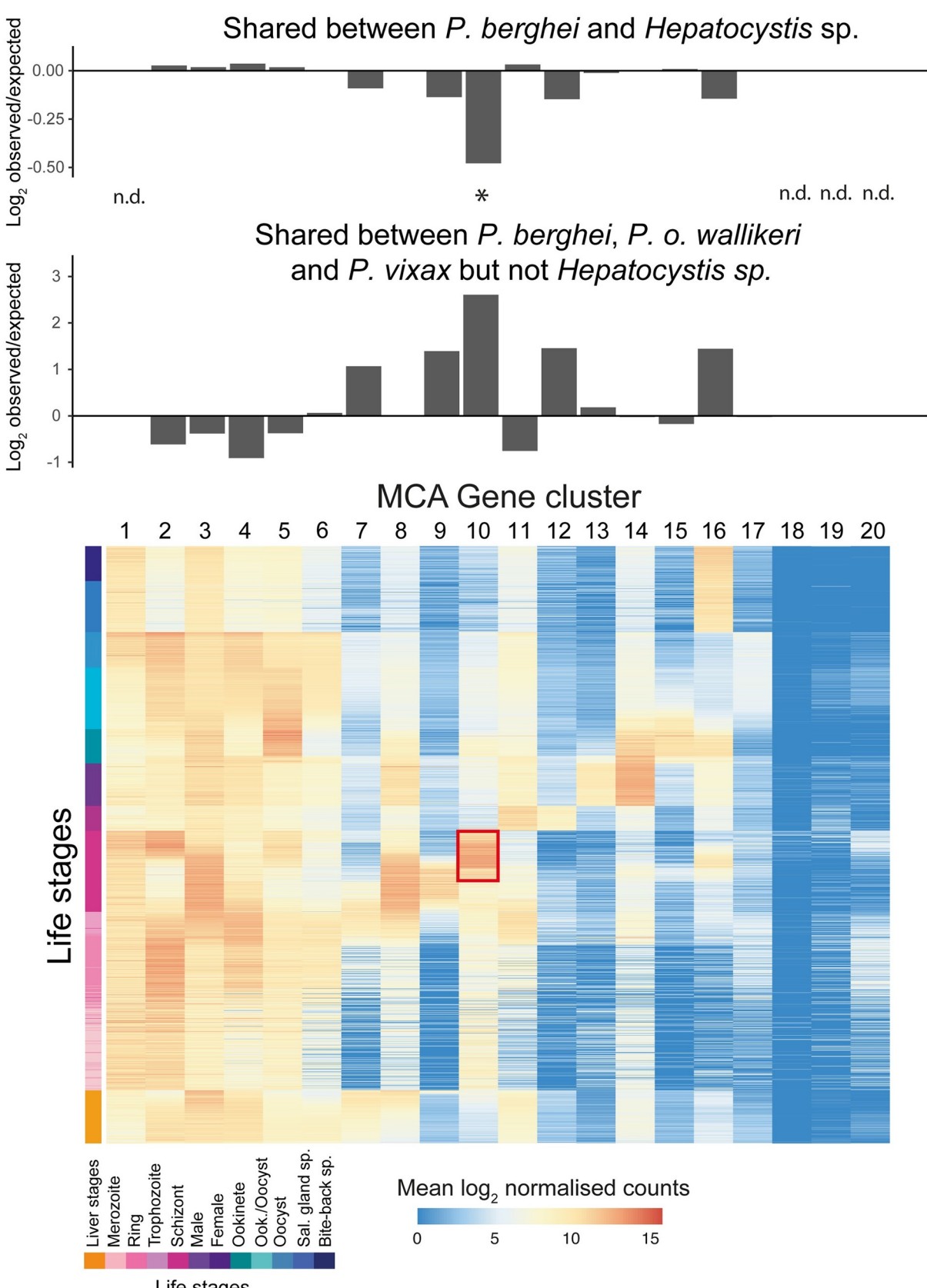

**Fig 4. *Hepatocystis* sp. orthologues of genes which are highly expressed in late schizogony in *P. berghei* tend to be missing from the genome.** *P. berghei* gene expression clusters from the Malaria Cell Atlas were used to determine whether orthologous genes, conserved with *P. vivax* and *P. ovale*, but absent in *Hepatocystis*, tended to be expressed in particular parts of the life cycle. The only significant cluster was cluster 10, which includes genes most highly expressed in late schizont stages (highlighted by a red box). The top panels show the $\log_2$ observed/ expected ratios for orthologous genes in each cluster which are shared between *P. berghei* and *Hepatocystis* and the same ratio for those which are shared between *P. berghei*, *P. vivax* and *P. ovale*, but not *Hepatocystis* sp. Cluster 1, 18, 19 and 20 were not tested because they contained fewer than 2 expected counts for either ratio. The asterisk indicates a Fisher's exact test false discovery rate of $< = 0.05$.

mutations (dN) rather than the more commonly used dN/dS. We first looked for enrichment of conserved protein domain families in genes with the highest 1% of dN values. There was an enrichment for the AP2 domain (Pfam:PF00847.20; Fisher test with BH correction; p-value = 0.01). Most *Plasmodium* species possess 27 ApiAP2 transcription factors containing this domain, which are thought to be the key players in control of gene expression and parasite development across the life cycle. AP2-G plays an important role in exiting the cycle of schizogony and commitment to gametocytogenesis in *Plasmodium* spp. [26,27], whereas, as demonstrated, *Hepatocystis* sp. lacks erythrocytic schizogony. *Hepatocystis* spp. also form much larger cysts in the liver (giving the genus its name) and develop in different tissues within a different insect vector compared to *Plasmodium* spp. Our *Hepatocystis* sp. assembly contained orthologues of all 27 ApiAP2 genes present in *P. falciparum* (Table 2). This suggests that life cycle differences between *Plasmodium* and *Hepatocystis* spp. are not reflected in gain or loss of these key transcription factors. However, the relatively high rate of non-synonymous mutations suggests there may have been significant adjustment in how these transcription factors act. To determine parts of the life cycle that were enriched for the most rapidly evolving genes, we looked at whether particular gene expression clusters from the Malaria Cell Atlas [19,20] were enriched for genes with high dN (top 5% of values; Table 3; S12A Fig; S4 Table). We found that three clusters (2, 4 and 6) had fewer genes with high dN than expected by chance (Fisher's exact test with Holm multiple hypothesis testing correction, p-value < 0.05) and that these contained genes expressed across much of the life cycle, especially growth phases. These clusters also tended to contain essential genes expected to be highly conserved in *Hepatocystis* sp. Although there was not a significant trend for gametocyte-associated genes having higher than average dN, the top 5% of genes ranked by dN contained several putative gametocyte genes (Table 3). Two of these encode putative 6-cysteine proteins P47 and P38, the first required for female gamete fertility [28]. Additionally, Merozoite TRAP-like protein (MTRAP), essential for *Plasmodium* gamete egress from erythrocytes [29] and two genes (HEP_00254800 and HEP_00195400) with orthologues involved in osmiophilic body formation [30,31] had high dN values. Overall, clusters involved in ookinete (15) and general mosquito stages (16) had significantly higher values than other clusters (Kolmogorov-Smirnov test: Cluster 15 vs all other clusters: D = 0.42, p-value = 1.05e-05. Cluster 16 vs all other clusters: D = 0.52, p-value = 4.50e-12; S12B Fig). This is also reflected in the *Hepatocystis* sp. genes with the highest dN values, which include oocyst rupture protein 2 (*orp2*; dN = 1.08), *ap2-o*, *ap2-sp2*, secreted ookinete protein (*psop7*) and osmiophilic body protein (*g377*). These genes provide clues about changes in the parasite that might relate to its adaptation to transmission by biting midges, rather than mosquitoes.

## Discussion

We have taken advantage of parasite reads captured as part of a primate genome sequencing study in order to assemble and annotate a draft quality genome sequence for a species of *Hepatocystis*. Our nuclear and apicoplast genome phylogenies confirm the recently proposed phylogenetic placement of this genus as an outgroup to the rodent-infecting *Vinckeia* subgenus of

**Table 3. Top 15 genes with functional annotations ranked by *Hepatocystis* sp. dN in comparison of *Hepatocystis* sp., *P. berghei* ANKA and *P. ovale curtisi*.** Genes with completely unknown function and genes with very little information on their possible functions have been left out from this table. The rank column indicates the *Hepatocystis* sp. dN rank of each gene in the complete table (with 4009 genes) that includes genes with unknown function (S4 Table).

| Gene id | Hepato-cystis dN | P. berghei dN | P. ovale dN | Annotations | Rank | Putative function |
|---------|------------------|---------------|-------------|-------------|------|-------------------|
| HEP_00146800 PBANKA_1303400 PocGH01_12075300 | 1.08 | 0.21 | 0.42 | Oocyst rupture protein 2 (ORP2) | 3 | Sporozoite egress from the oocyst [32] |
| HEP_00446500 PBANKA_1003000 PocGH01_03012800 | 0.71 | 0.19 | 0.34 | liver specific protein 2 (LISP2) | 19 | Involved in liver stage development [33] |
| HEP_00295100 PBANKA_0905900 PocGH01_09049300 | 0.71 | 0.31 | 0.63 | AP2-O | 20 | Essential for morphogenesis in ookinete stage in *Plasmodium* [34] |
| HEP_00035800 PBANKA_1107600 PocGH01_10033500 | 0.69 | 0.27 | 0.16 | 6-cysteine protein (p38) | 24 | *P. berghei* p38 is expressed in gametocytes and in asexual blood stages [28] |
| HEP_00213800 PBANKA_1001800 PocGH01_03011500 | 0.67 | 0.45 | 0.63 | AP2 domain transcription factor AP2-SP2 | 28 | Required for sporozoite production in *Plasmodium [35]* |
| HEP_00337100 PBANKA_0112100 PocGH01_11043100 | 0.62 | 0.36 | 0.42 | AP2 domain transcription factor ApiAP2 | 40 | Involved in blood stage replication [35,36] |
| HEP_00456700 PBANKA_0512800 PocGH01_06021700 | 0.62 | 0.61 | 0.38 | Merozoite TRAP-like protein (MTRAP) | 42 | Essential for gamete egress from erythrocytes [29] |
| HEP_00304800 PBANKA_1353400 PocGH01_12023100 | 0.62 | 0.32 | 0.26 | Secreted ookinete protein (PSOP7) | 43 | Secreted ookinete proteins are necessary for invasion of the mosquito midgut [37] |
| HEP_00254800 PBANKA_1449000 PocGH01_14060000 | 0.59 | 0.26 | 0.26 | Microgamete surface protein MiGS | 54 | Plays a critical role in male gametocyte osmiophilic body formation and exflagellation [30] |
| HEP_00115700 PBANKA_0304400 PocGH01_04023000 | 0.55 | 0.41 | 0.15 | Merozoite surface protein 4 (MSP4) | 62 | Merozoite surface proteins are involved in red blood cell invasion [38] |
| HEP_00166600 PBANKA_0301000 PocGH01_04026800 | 0.54 | 0.36 | 0.2 | Repetitive organellar protein (ROPE) | 64 | Localised to the apical end of merozoites, possibly involved in red blood cell invasion [39] |
| HEP_00195400 PBANKA_1463000 PocGH01_14074600 | 0.54 | 0.24 | 0.26 | Osmiophilic body protein (G377) | 67 | Female-specific protein, affects the size of the osmiophilic body and female gamete egress efficiency [31] |
| HEP_00130400 PBANKA_1358000 PocGH01_12018000 | 0.53 | 0.17 | 0.11 | Thioredoxin 2 (TRX2) | 75 | Part of a protein complex in parasitophorous vacuolar membrane, required for pathogenic protein secretion into host [40], important for maintaining normal blood-stage growth [41] |
| HEP_00391300 PBANKA_0313400 PocGH01_04013500 | 0.52 | 0.49 | 0.27 | Autophagy-related protein 11 (ATG11) | 76 | Predicted to be involved in cargo selection in selective autophagy [42] |
| HEP_00155500 PBANKA_1302300 PocGH01_12076400 | 0.52 | 0.18 | 0.17 | Metacaspase-2 | 80 | Protease with caspase-like activity [43] |

*Plasmodium [2]*. However, a distinct branching pattern and low bootstrap support for many nodes in our mitochondrial genome phylogeny highlights why some previous analyses have come to different conclusions about the placement of *Hepatocystis*. Thus, the use of mitochondrial genes to infer phylogenetic relationships between species within the Haemosporidia should be approached with caution. We found a long branch leading to *Hepatocystis* sp.,

suggesting a relatively deep split from the rodent-infecting species. In addition, we showed robustly that *Hepatocystis* sp. clusters, as expected, with *Hepatocystis epomophori*, which infects bats. This finding supports the polyphyly of the *Plasmodium* parasites infecting apes, monkeys and rodents but the monophyly of *Hepatocystis* itself. A close relative of rodent-infecting *P. berghei* (*P. cyclopsi*) has been found in bats [10] and thus the *Hepatocystis*/*Vinckeia* group represents a relatively labile group with respect to host preference. Indeed, the possibility of cross-species transmission of *Hepatocystis* sp. was reported previously [6].

The paraphyly of *Plasmodium* with respect to *Hepatocystis* exists because *Hepatocystis* spp. lack a defining characteristic of the *Plasmodium* genus, namely erythrocytic schizogony—asexual development in the blood. Thus, the very part of the *Plasmodium* life cycle which causes the symptoms of malaria is thought to be absent in *Hepatocystis*. While multiple lines of enquiry have failed to identify these forms (Garnham, 1966) there has remained some doubt, with reports of cells with schizont-like morphology in the blood for some species [13,44]. We were able to take advantage of bulk RNA-seq data collected from blood samples of a number of red colobus monkeys, all apparently infected with very closely related *Hepatocystis* parasites. By comparing to single-cell RNA-seq data from known cell types, we found no evidence for schizont stages in the blood. The apparent lack of schizonts could be due to sequestration of these stages away from the bloodstream. This is seen in some *Plasmodium* species, so we cannot rule out schizogony away from peripheral circulation. However, in its current state, this lack of evidence supports the idea that erythrocytic schizogony has been lost three times: in *Hepatocystis*, *Polychromophilus* and *Nycteria* [2]. Ancestrally, gametocytogenesis must have been the default developmental pathway, being required for transmission. However, in *Plasmodium*, it seems that erythrocytic schizogony became the default developmental pathway with epigenetic control of the ApiAP2-G transcription factor, required for development into sexual stages [26,27]. Perhaps the simplest explanation for a loss of erythrocytic schizogony would be that ApiAP2-G is no longer under control, but is constitutively expressed in parasites leaving the liver. In line with this idea we find ApiAP2-G present and highly expressed in blood stage *Hepatocystis* sp. Furthermore, all *Plasmodium* ApiAP2 transcription factors are conserved in *Hepatocystis* sp., indicating that changes in its life cycle are not associated with their loss or gain.

The lack of erythrocytic schizogony is supported by a tendency for orthologous genes missing in *Hepatocystis* sp. (relative to *Plasmodium* spp.) to be those expressed in blood schizonts. The most noticeable example is the complete absence of the Reticulocyte Binding Protein (RBP) family, found across all *Plasmodium* spp. examined so far, including those which infect birds [45]. RBP proteins are known to function as essential red blood cell invasion ligands in *Plasmodium falciparum* [46] and multiple copies are thought to provide alternative invasion pathways [25]. However, previous transcriptomic data have suggested that, while *rbp* genes in *P. berghei* are highly expressed in schizonts, they are less abundant or have a distinct repertoire in liver stages [20,47,48]. This implies that RBPs are less important, or at least that distinct invasion pathways are used by first generation merozoites. Also missing were *cdpk5* (involved in schizont egress) and *msp9* (an invasion related gene enriched in blood vs. liver schizonts in Caldelari et al. [47]). Taken together, these results underscore the increasing realisation that first generation merozoites have distinct properties from merozoites that have developed in the blood and suggest the way in which first generation merozoites invade red blood cells may be distinct. Looking more widely across the *Haemosporidia*, we found scant evidence for RBPs outside of the *Plasmodium* genus. There were no matches at all in the draft genome sequence of the relatively close *Plasmodium* outgroup *Haemoproteus tartakovskyi*. However, there were fragmentary matches to RBPs from *Plasmodium* species infecting birds in transcriptome assemblies of *Hae. columbae* and the more distant *Leucocytozoon buteonis*. The *Hae. columbae*

fragment aligned to a conserved region of this divergent family, suggesting it could encode some core function of RBPs. More complete genome sequences from across the haemosporidians will be needed to understand the evolution of this gene family, which seems to be key for understanding the host specificity of *Plasmodium* parasites [49].

The genomes of *Plasmodium* spp. each contain large, rapidly evolving gene families that are known, or thought to be involved in host-parasite interactions, principally in asexual blood stages. The reason for their numbers may be due to a bet-hedging strategy, providing the diversity necessary for evading adaptive immune responses or dealing with unpredictable host variation. Although the *Hepatocystis* sp. genome contains two novel multigene families, we identified only 10–15 copies of each. The largest gene families in the closely related rodent-infecting *Plasmodium* species (*pir* and *fam-a*) are present here only as their ancestral orthologues, also conserved in the monkey-infecting *Plasmodium* species. We should be cautious in noting a lack of expansion in such families in *Hepatocystis* sp., as previous draft *Plasmodium* genome sequences have been shown to under-represent these genes. However, it is perhaps not surprising that *pir* genes are poorly represented. They are thought to play a role in the maintenance of chronic infections mediated by asexual stages in the blood [18] and *Hepatocystis* infection does not involve this stage of development. Given that *Hepatocystis* can sustain long chronic infections [1], presumably from the liver, this parasite may help us to better understand how *Plasmodium* survives in the liver.

A striking feature of the *Hepatocystis* life cycle is its vector—a biting midge rather than a mosquito. We found evidence of rapid evolution amongst orthologues of *Plasmodium* genes involved in mosquito stages of development, suggesting that adaptation to a new insect vector was a major evolutionary force. These rapidly evolving genes provide insights into parasite-vector interactions and may provide avenues for the development of interventions to prevent transmission of the malaria parasite.

Overall, our findings demonstrate the insights that can be gained into malaria parasite biology from relatives of *Plasmodium*, even with draft-quality genome sequences. In the future, we expect that high-quality genome sequences of *Hepatocystis* spp., and additional relatives from genera such as *Nycteria*, *Haemoproteus*, and *Polychromophilus*, will be of great value for understanding the evolution and molecular biology of one of humanity's greatest enemies.

## Methods

### Ethics statement

All animal research was approved by the Uganda Wildlife Authority (permit number UWA/TDO/33/02), the Uganda National Council for Science and Technology (permit number HS 364), and the University of Wisconsin-Madison Animal Care and Use Committee (V005039) prior to initiation of the study. Biological materials were shipped internationally under CITES permit #002290 (Uganda). The animal was anesthetized with a combination of Ketamine (5 mg/kg) and Xylazine (2 mg/kg) administered intramuscularly using a variable-pressure air rifle (Pneudart, Inc, Williamsport, PA, USA). After sampling, the animal was given a reversal agent (Atipamezole, 0.5 mg/kg), and released after recovery back to its social group. All animal use followed the guidelines of the Weatherall Report on the use of non-human primates in research.

### Sample collection and data generation

The sequence data used in this study were part of a project originally designed to generate a reference genome for the red colobus monkey (genus *Piliocolobus*). Biomaterials used were from wild Ugandan (or Ashy) red colobus monkey (*Piliocolobus tephrosceles*) individuals from

Kibale National Park, Uganda. These animals reside in a habituated group that has been a focus of long-term studies in health, ecology, and disease [12,50,51]. Red colobus individuals were immobilised in the field as previously described [52]. Whole blood was collected using a modified PreAnalytiX PAXgene Blood RNA System protocol as described in Simons et al. [12]. Additionally, whole blood was collected into BD Vacutainer Plasma Preparation Tubes, blood plasma and cells were separated via centrifugation, and both were subsequently aliquoted into cryovials and stored in liquid nitrogen. Samples were transported to the United States in an IATA-approved liquid nitrogen dry shipper and then transferred to −80 ˚C for storage until further processing.

Methods for DNA extraction, library preparation, and whole genome sequencing are described in Simons [53]. Briefly, high molecular weight DNA was extracted from the blood cells of one red colobus monkey individual and size selected for fragments larger than 50,000 base pairs. A 10X Genomics Chromium System library preparation was performed and subsequently sequenced on two lanes of a 150 bp paired-end Illumina HiSeqX run as well as two lanes of a 150 bp paired-end Illumina HiSeq 4000 run.

Methods for RNA extraction and library preparation are described in Simons et al. (2019). Briefly, RNA was extracted from 29 red colobus individuals using a modified protocol for the PreAnalytiX PAXgene Blood RNA Kit protocol. Total RNA extracts were concentrated, depleted of alpha and beta globin mRNA, and assessed for integrity (RIN mean: 8.1, range: 6.6–9.2). Sequencing libraries were prepared using the KAPA Biosystems Stranded mRNA-seq Kit and sequenced on four partial lanes of a 150 bp paired-end Illumina HiSeq 4000 run. These data were uploaded to NCBI as part of BioProject PRJNA413051.

## Separation of *Hepatocystis* and *Piliocolobus* scaffolds

The *Piliocolobus tephrosceles* genome assembly (ASM277652v1) was downloaded from the NCBI database. Scaffolds were first sorted by their GC% and Diamond 0.9.22 [54] BLASTX hits against a database of representative apicomplexan and Old World monkey proteomes. The sorting was improved by examining mapping scores of the scaffolds mapped to *Plasmodium* species and *Macaca mulatta* genomes (Mmul_8.0.1, GenBank assembly accession GCA_000772875.3) using Minimap2 2.12 [55]. The separation of scaffolds was further verified and refined by running NCBI BLAST of 960 bp fragments of all scaffolds against the NCBI nt database (Jul 18 2017 version) [56]. To predict genes in the apicomplexan scaffolds, Companion automatic annotation software [7] was run with these scaffolds as input and the *P. vivax* P01 genome as the reference.

## Identification of *Hepatocystis* sequences in Piliocolobus RNA-seq data

Illumina HiSeq 4000 RNA-seq reads from the study PRJNA413051 were downloaded from the European Nucleotide Archive. In order to find out if the RNA-seq data contained apicomplexan sequences, mapping of these reads to apicomplexan scaffolds from *Piliocolobus tephrosceles* genome assembly (ASM277652v1) was done using HISAT2 2.1.0 [57].

## *Hepatocystis* genome assembly

**Filtering of reads for assembly.** Minimap2 [55] and Kraken 2.0.8-beta [58] were used to identify the best matching species for each 10x Chromium genomic DNA read (from Illumina HiSeq X and HiSeq4000 platforms). Our Kraken database contained 17 Old World monkey genomes and 19 *Plasmodium* genomes downloaded from NCBI FTP in June 2018 [56]. The Kraken database also included the contigs of the *P. tephrosceles* assembly ASM277652v1, separated into *P. tephrosceles* and *Hepatocystis* sp. *Plasmodium malariae* UG01 (from PlasmoDB

[59] version 39) and *Macaca mulatta* (Mmul_8.0.1) assemblies were used as reference genomes for the assignment of reads based on Minimap2 mapping scores. Reads that were unambiguously identified as monkey sequences using Kraken and Minimap2 were excluded from subsequent assemblies. The Supernova assembler manual [60] warns against exceeding 56x coverage in assemblies. Reads selected for Supernova assemblies were therefore divided into 34 batches, with ~10 million reads in each batch. Reads were ordered by their barcodes so that those with the same barcode would preferentially occur in the same batch.

**Supernova and SPAdes assemblies.** We generated 34 assemblies with Supernova v2.1.1 with default settings. In addition, two SPAdes v3.11.0 [61] assemblies with default settings were generated with *Hepatocystis* reads: one with HiSeq X reads and another with HiSeq 4000 reads. Chromium barcodes were removed from the reads before the SPAdes assemblies.

**Deriving the mitochondrial sequence.** *Hepatocystis* Supernova and SPAdes assembly contigs were mapped to the *P. malariae* UG01 genome from PlasmoDB version 40 with Minimap2. The sequences of contigs that mapped to the *P. malariae* mitochondrion were extracted using SAMTools 0.1.19-44428cd [62] and BEDtools v2.17.0 [63]. The contigs were oriented and then aligned using Clustal Omega 1.2.4 [64]. Consensus sequence of aligned contigs was derived using Jalview 2.10.4b1 [65]. The consensus sequence was circularised with Circlator minimus2 [66].

**Canu assembly.** Scaffolds from the Supernova assemblies were broken into contigs. All contigs from the Supernova and SPAdes assemblies were pooled and used as the input for Canu assembler 1.6 [67] in place of long reads. Canu assembly was done without read correction and trimming stages. The settings for Canu were as follows: -assemble genomeSize = 23000k minReadLength = 300 minOverlapLength = 250 corMaxEvidenceErate = 0.15 correctedErrorRate = 0.16 stopOnReadQuality = false -nanopore-raw.

**Processing of Canu unassembled sequences file.** Selected contigs from the Canu unassembled sequences output file (*.unassembled.fasta) were recovered and pooled with assembled contigs (*.contigs.fasta). The first step in the filtering of the contigs of the unassembled sequences file was to exclude contigs that had a BLAST match in the assembled sequences output file (with E value cutoff 1e-10). Next, contigs where low complexity sequence content exceeded 50% (detected using Dustmasker 1.0.0 [68]) were removed. Contigs with GC content higher than 50% were also removed. Diamond BLASTX (against a database of *Macaca mulatta*, *P. malariae* UG01, *P. ovale wallikeri*, *P. falciparum* 3D7 and *P. vivax* P0 proteomes) and BLAST (using the nt database from Jul 18 2017 and nr database from Jul 19 2017) were then used to exclude all contigs where the top hits were not an apicomplexan species. In total, 0.34% of contigs from the unassembled sequences file were selected to be included in the assembly.

**Deduplication of contigs.** Initial deduplication of contigs was done using BBTools dedupe [69] (Nov 20, 2017 version) and GAP5 v1.2.14-r3753M [70] autojoin. In addition, BUSCO 3.0.1 [71] was used to detect duplicated core genes with the protists dataset. Two contigs flagged by BUSCO as containing duplicated genes were removed. All vs all BLAST of contigs (with E-value cutoff 1e-20, minimum overlap length 100 bp, minimum identity 85%) was used to find possible cases of remaining duplicated contigs. Contigs yielding BLAST hits were aligned with MAFFT v7.205 [72] and the alignments were manually inspected. Contained contigs were deleted and contigs that had unique overlaps with high identity were merged into consensus sequences using Jalview.

**Removal of contaminants after Canu assembly.** All Canu assembly contigs were checked with Diamond against a database of *Macaca mulatta*, *P. malariae* UG01, *P. ovale wallikeri*, *P. falciparum* 3D7 and *P. vivax* P01 proteomes. The Diamond search did not detect any contaminants. Contigs not identified by Diamond were checked with BLAST against the nt database

(Jul 18 2017 version). Contigs where the top BLAST hit was a human or monkey sequence were removed from the assembly.

A subset of contigs in the assembly was observed to consist of short sequences with low complexity, high GC% and low frequency of stop codons. These contigs did not match any sequences by BLAST search against nt and nr databases (with E-value cutoff 1e-10). Due to their difference from the rest of the contigs in the assembly, it was assumed that these contigs were contaminants rather than *Hepatocystis* sequences. In order to programmatically find these contigs, GC%, tandem repeats percentage, percentage of low complexity content and frequency of stop codons were recorded for all contigs in the assembly. Tandem Repeats Finder 4.04 [73] was used to assess tandem repeats percentage and Dustmasker 1.0.0 [68] was used to find low complexity sequence content. PCA and k-means clustering (using R version 3.5.1) showed that the assembly contigs separated into two groups based on these parameters. The group of contigs with low complexity (189 contigs) was removed from the assembly.

**Scaffolding and polishing of Canu assembly contigs.** Before scaffolding, contigs were filtered by size to remove sequences shorter than 200 bp. *Hepatocystis* RNA-seq reads were extracted from RNA-seq sample SAMN07757854 using Kraken 2. Canu assembly contigs were scaffolded with these reads using P_RNA_scaffolder [74]. To correct scaffolding errors, the scaffolds were processed with REAPR 1.0.18 [75] using 197819014 unbarcoded *Hepatocystis* DNA read pairs. REAPR was run with the *perfectmap* option and -break b = 1. Next, the assembly was scaffolded using Scaff10x (https://github.com/wtsi-hpag/Scaff10X) version 3.1, run for 4 iterations with the following settings: -matrix 4000 -edge 1000 -block 10000 -long-read 0 -link 3 -reads 5. 197,819,014 *Hepatocystis* DNA read pairs were used for Scaff10x scaffolding. After this, P_RNA_scaffolder was run again as above. This was followed by running Tigmint 1.1.2 [76] with 419,652,376 *Hepatocystis* read pairs to correct misassemblies. fill_-gaps_with_gapfiller (https://github.com/sanger-pathogens/assembly_improvement/blob/master/bin/fill_gaps_with_gapfiller) was used to fill gaps in scaffolds, using 197819014 unbarcoded *Hepatocystis* DNA read pairs. After this, ICORN v0.97 [77] was run for 5 iterations with 4608740 *Hepatocystis* read pairs. This was followed by polishing the assembly with Pilon 1.19 [78] using 21,794,613 *Hepatocystis* read pairs. Assembly completeness was assessed with CEGMA v2.5 [79]. *P. berghei* ANKA, *P. ovale curtisi* and *P. falciparum* 3D7 genomes from PlasmoDB release 45 were also assessed with CEGMA with the same settings in order to compare the *Hepatocystis* assembly with *Plasmodium* assemblies.

## Curation and annotation of the Hepatocystis genome assembly

The assembly was annotated using Companion [7]. The alignment of reference proteins to target sequence was enabled in the Companion run but all other parameters were left as default. A GTF file derived from mapping of *Hepatocystis* RNA-seq reads of three biological samples (SAMN07757854, SAMN07757861 and SAMN07757872) to the assembly was used as transcript evidence for Companion. To produce the GTF file, the RNA-seq reads were mapped to the assembly using 2-pass mapping with STAR RNA-seq aligner [80] (as described in the "Variant calling of RNA-seq samples" section) and the mapped reads were processed with Cufflinks [81]. All *Plasmodium* genomes available in the web version of Companion were tested as the reference genome for annotating the *Hepatocystis* genome, in order to find out which reference genome yields the highest gene density. For the final Companion run the *P. falciparum* 3D7 reference genome (version from June 2015) was used. The Companion output was manually curated using Artemis [82] and ACT [83] version 18.0.2. Manual curation was carried out to correct the overprediction of coding sequences, add missing genes and correct exon-intron boundaries. Altogether 680 gene models were corrected, 546 genes added and 221 genes

deleted. RNA-seq data was used as supporting evidence. Non-coding RNAs were predicted with Rfam [84].

All genes were analysed for the presence of a PEXEL-motif using the updated HMM algorithm ExportPred v2.0 [85]. Distant homology to *hep1* and *hep2* gene families was sought by using the HHblits webserver with default options [14].

The reference genomes used to produce statistics on features of *Plasmodium* genomes in Fig 2 and Table 1 were as follows: *P. relictum* SGS1, *P. gallinaceum* 8A [45], *P. malariae* UG01, *P. ovale wallikeri*, *P. ovale curtisi* GH01 [11], *P knowlesi* H [86], *P. vivax* P01 [87], *P. cynomolgi* M [88], *P. chabaudi* AS [18], *P. berghei* ANKA [89], *P. reichenowi* CDC [90], *P. falciparum* 3D7 [91].

For S1 Table, transmembrane domains of proteins were predicted using TMHMM 2.0 [92]. Conserved domains were detected in proteins using HMMER i1.1rc3 (http://hmmer.org/) and Pfam-A database release 28.0 [93], with E-value cutoff 1e-5. Besides predicting exported proteins with ExportPred 2 [85], matches to PEXEL consensus sequence (RxLxE/Q/D) were counted in protein sequences using string search in Python. Signal peptides were detected using SignalP-5 [94].

## Analysis of other Haemosporidian genomes and transcriptomes

**Genomes and transcriptomes of other Haemosporidians.** A *Haemoproteus tartakovskyi* genome assembly was downloaded from the Malavi database (http://130.235.244.92/Malavi/Downloads/Haemoproteus_tartakovskyi). The Companion annotation tool [7] was used to automatically annotate the assembly, using the *P. falciparum* 3D7 genome as the reference. The alignment of reference proteins to target sequence was enabled and the rest of the settings were left as default. The genome annotation was further edited manually using Artemis 18.1.0 [82].

A transcriptome assembly of *Haemoproteus columbae* [23] was downloaded from GenBank (GenBank ID GGWD00000000.1). Illumina MiSeq reads of *Leucocytozoon* and its host *Buteo buteo* were downloaded from the European Nucleotide Archive (study PRJEB5722). The reads from all four samples were processed with Cutadapt 2.7 (http://journal.embnet.org/index.php/embnetjournal/article/view/200) to remove artificial Illumina sequences and then assembled with SPAdes v3.13.1 [95] with the—rna flag. Contigs from the transcripts.fasta and soft_filtered_transcripts.fasta output files of SPAdes were pooled. *Leucocytozoon* contigs were separated from the contigs of *Buteo buteo* and bacterial contaminants using Diamond 0.9.24 [54], BLAST and Minimap2 2.17-r941 [55], similarly with how *Hepatocystis* contigs were isolated from the *Piliocolobus* assembly ASM277652v1. GAP5 1.2.14-r [70] was used to join the *Leucocytozoon* assembly contigs by unique overlaps. The *Leucocytozoon* assembly contigs were then polished using Pilon 1.23 [78] (3 iterations).

The completeness of the *Haemoproteus* and *Leucocytozoon* assemblies was assessed using CEGMA 2.5 [79]. The CEGMA completeness statistics for these Haemosporidian assemblies were the following. *Haemoproteus tartakovskyi* genome assembly: Completeness Complete: 64.52%, Completeness Partial: 68.55%. *Haemoproteus columbae* transcriptome assembly: Completeness Complete: 37.90%, Completeness Partial: 57.26%. *Leucocytozoon buteonis* transcriptome assembly: Completeness Complete 25.40%, Completeness Partial 35.08%.

**BLAST searches for RBPs.** A BLAST database was made from Plasmodium RBPs downloaded from PlasmoDB [96] (release 46). NCBI blastx 2.9.0+ with e-value cutoff 1e-5 was run against this database, using the *Haemoproteus* genome and transcriptome assemblies and the *Leucocytozoon* transcriptome assembly as queries. The transcripts that yielded BLAST hits were further examined using BLAST against the NCBI nt and nr databases (April 2020 versions) and by sequence alignments with RBPs from PlasmoDB. One of the *Haemoproteus* transcripts

(GGWD01016989.1) matched RBPs in the blastx search (the best match was with PRELSG_0 014300: E-value 1.31e-23, score: 89.0) and did not yield non-RBP BLAST hits in searches against the nt and nr databases. One *Leucocytozoon* transcript also matched *Plasmodium* RBPs (top match: PRELSG_0013000, E-value 1.06e-09, score: 56.2). The sequence of the *Haemoproteus* transcript GGWD01016989.1 was translated to amino acids using ExPASy Translate (April 2020 version) [97] and then aligned with a selection of *Plasmodium* RBPs from PlasmoDB (release 46) using MAFFT 7 [98] with default settings. The resulting alignment was cropped in Jalview 2.10.4b1 to include only the region that contained the *Haemoproteus* sequence [99]. A phylogenetic tree was generated from the alignment as described in the next section.

## Phylogenetic trees

Haemosporidian sequences were downloaded from NCBI FTP and PlasmoDB (release 43). The phylogenetic tree of cytochrome B and the tree that included 11 *Hepatocystis epomophori* genes were based on DNA alignments. The cytochrome B tree also included cytochrome B sequences from *de novo* assemblies of *Hepatocystis* RNA-seq reads derived from *Piliocolobus tephrosceles* blood. The trees of mitochondrial, apicoplast and nuclear proteomes were based on protein alignments. For apicoplast proteome and nuclear proteome trees, orthologous proteins were identified using OrthoMCL 1.4 [100]. The OrthoMCL run included the Haemoproteus tartakovskyi proteome that had been derived from the *Haemoproteus tartakovskyi* genome assembly using Companion, as previously described. All vs all BLAST for OrthoMCL was done using blastall 2.2.25 with E-value cutoff 1e-5. OrthoMCL was run with mode 3. Proteins with single copy orthologs across all the selected species were used for the protein phylogenetic trees. Sequences were aligned with MAFFT 7.205 [98] (with—auto flag) and the alignments were processed using Gblocks 0.91b [101] with default settings. Individual Gblocks-processed alignments were concatenated into one alignment. The phylogenetic trees were generated using IQ-TREE multicore version 1.6.5 [102] with default settings and plotted using FigTree 1.4.4 (https://github.com/rambaut/figtree/releases). Inkscape (https://inkscape.org) version 0.92 was used to edit text labels of the phylogenetic trees generated with FigTree.

## Clustering of *pir* proteins into subfamilies

Sequences of *Plasmodium pir* family proteins (including *bir*, *cyir*, *kir*, *vir* and *yir* proteins) were downloaded from PlasmoDB [59] (release 39). The sequences were clustered using MCL [103], following the procedures described in the section "Clustering similarity graphs encoded in BLAST results" in clmprotocols (https://micans.org/mcl/man/clmprotocols.html). The BLAST E-value cutoff used for clustering was 0.01 and the MCL inflation value was 2. The *pir* protein counts per subfamily in each species were plotted as a heatmap using the heatmap.2 function in gplots package version 3.0.1.1 in R version 3.5.1.

## Mapping and assembly of *Hepatocystis* sp. RNA-seq data

To separate *Hepatocystis* reads from *Piliocolobus* reads, RNA-seq data from the ENA (study PRJNA413051) were mapped to a FASTA file containing genome assemblies of *Hepatocystis* and *M. mulatta* (NCBI assembly Mmul_8.0.1), using HISAT2 version 2.1.0 [57], with "—rnastrandness RF". BED files were generated from the mapped reads using BEDTools 2.17.0 [63]. Reads from each technical replicate were merged, resulting in a single set of read counts for each individual monkey. The BED files were filtered to remove multimapping reads and reads with mapping quality score lower than 10. Names of reads that specifically mapped to the *Hepatocystis* assembly were extracted from the BED file. SeqTK 1.0-r31 (https://github.com/lh3/seqtk) was used to isolate *Hepatocystis* FASTQ reads based on the list of reads from the

previous step. The *Hepatocystis* reads were then mapped to the *Hepatocystis* genome assembly using HISAT2 2.1.0 with "—rna-strandness RF" flag. The SAM files with mapped reads were converted to sorted BAM files with SamTools 0.1.19-44428cd [62]. The EMBL file of *Hepatocystis* genome annotations was converted to GFF format using Artemis 18.0.1 [82]. Htseq-count 0.7.1 [104] was used to count mapped reads per gene in the GFF file with "-t mRNA -a 0 -s reverse". Htseq-count files of individual RNA-seq runs were merged into a single file.

In order to extract *Hepatocystis* cytochrome b sequences of each RNA-seq sample, *Hepatocystis* RNA-seq reads of each sample were isolated from *Piliocolobus* reads as described above and then assembled with the SPAdes assembler v3.11.0 [105] with the "—rna" flag. *Hepatocystis* cytochrome b contigs were identified in each of the 29 RNA-seq assemblies using BLAST against *Hepatocystis* cytochrome b from the DNA assembly (E-value cutoff 1e-10).

In addition to assemblies of individual RNA-seq samples, an assembly of all RNA-seq samples pooled was done. The reads for this assembly were sorted by competitive mapping to *P. ovale curtisi* GH01 (from PlasmoDB release 45) and *Macaca mulatta* (Mmul_8.0.1, GenBank assembly accession GCA_000772875.3) genomes with Minimap2 (with the "-ax sr" flag). Reads mapping to the *Macaca mulatta* genome with minimum mapping score 20 were removed and the rest of the reads were assembled with the SPAdes assembler v3.13.1 [105] with the "—rna" flag. *Hepatocystis* contigs were identified by comparison of sequences with *Plasmodium* and *Macaca mulatta* reference genomes using Diamond, Minimap2 and BLAST, similarly to what is described in the section "Separation of *Hepatocystis* and *Piliocolobus* scaffolds". Further decontamination was done using Diamond and BLAST searches against 19747 sequences from *Ascomycota* and 165860 bacterial sequences downloaded from UniProt (release 2019_10) [106] and 3 *Babesia* proteomes from PiroplasmaDB (release 46) [107]. Selected contigs were also checked with BLAST against the NCBI nt database. The assembly was deduplicated using BBTools dedupe (Nov 20, 2017 version) and GAP5 v1.2.14-r3753M. Assembly completeness was assessed using CEGMA 2.5. In order to reduce the number of contigs so that they could be used as input for Companion, the assembly was scaffolded with RaGOO Version 1.1 [108], using the *Hepatocystis* DNA assembly as the reference. The assembly was then processed by the Companion annotation software (Glasgow server, November 2019 version, with *P. falciparum* 3D7 reference genome, with protein evidence enabled and the rest of the settings left as default). In order to detect proteins missed by Companion, EMBOSS Transeq (version 6.3.1) was used to translate the transcriptome assembly in all 6 reading frames. The output of Transeq was then filtered to keep sequences between stop codons with minimum length of 240 amino acids. Protein BLAST with E-value cutoff 1e-20 was used to detect sequences in Transeq output that were not present in the proteins annotated by Companion. These selected Transeq output sequences were checked for contaminants with BLAST similarly to what was described before. The sequences that passed the contaminant check were combined with the set of *Hepatocystis* RNA-seq assembly proteins that were detected by Companion. OrthoMCL was run with proteins from *Hepatocystis* RNA-seq assembly (Companion and selected Transeq sequences combined), *Hepatocystis* DNA assembly proteins, 20 *Plasmodium* proteomes from PlasmoDB release 43 and *P. ovale wallikeri* proteome (GenBank GCA_900090025.2). The settings for OrthoMCL were as described in the "Phylogenetic trees" section.

## dN analysis

*P. berghei* ANKA and *P. ovale curtisi* protein and transcript sequences were retrieved from PlasmoDB [59] (release 45). One-to-one orthologs between *Hepatocystis*, *P. berghei* ANKA and *P. ovale curtisi* were identified using OrthoMCL [100]and a Newick tree of the three species was generated with IQ-TREE [102]. The settings for OrthoMCL and IQ-TREE were as

described in the "Phylogenetic trees" section. Transcripts of one-to-one orthologs were aligned using command line version of TranslatorX [109] with "-p F -t T" flags, so that each alignment file contained sequences from three species. Gaps were removed from alignments while retaining the correct reading frame. Alignment regions where the nucleotide sequence surrounded by gaps was shorter than 42 bp were also removed. In addition, the script truncated alignments at the last whole codon if a sequence ended with a partial codon due to a contig break. The alignments and the Newick tree of the 3 species were then used as input for codeml [110] in order to determine the dN and dN/dS of each alignment. The codeml settings that differed from default settings were: seqtype = 1, model = 1. *P. berghei* RNA-seq cluster numbers from Malaria Cell Atlas [20] were assigned to each alignment based on the *P. berghei* gene in the alignment. Transcriptomics-based gametocyte specificity scores of *Plasmodium* genes were taken from an existing study on this topic [111] (transcripts S2 Table of "Transcriptomics_all_-studies" tab). The *P. falciparum* genes in the gametocyte specificity scores table were matched with equivalent *Hepatocystis* genes using OrthoMCL (run with the same settings as when used for phylogenetic trees). Statistical tests with the dN results (Kolmogorov Smirnov test, Fisher test and Spearman correlation) were performed using the *stats* library in R.

## Variant calling of RNA-seq samples

SNPs and indels were called in *Hepatocystis* RNA-seq reads that had been separated from *Piliocolobus* reads as described above. Four technical replicates of each RNA-seq sample were pooled. Variant calling followed the "Calling variants in RNAseq" workflow in GATK [112] user guide (https://gatkforums.broadinstitute.org/gatk/discussion/3891/calling-variants-in-rnaseq). First, the reads were mapped to the reference genome using 2-pass mapping with the STAR RNA-seq aligner [80] version 2.5.3a. 2-pass mapping consisted of indexing the genome with genomeGenerate command, aligning the reads with the genome, generating a new index based on splice junction information contained in the output of the first pass and then producing a final alignment using the new index. GATK [112] version 4.0.3.0 was used for the next steps. The mapped reads were processed with GATK MarkDuplicates and SplitNCigarReads commands. GATK HaplotypeCaller was then run with the following settings:—dont-use-soft-clipped-bases—emit-ref-confidence GVCF—sample-ploidy 1—standard-min-confidence-threshold-for-calling 20.0. Joint genotyping of the samples was then done using GATK CombineGVCFs and GenotypeGVCFs commands. This was followed by running VariantFiltration with these settings: -window 35 -cluster 3—filter-name FS -filter 'FS > 30.0'—filter-name QD -filter 'QD < 2.0'. SNPs were separated from indels using GATK SelectVariants. Samples SAMN07757853, SAMN07757863, SAMN07757870 and SAMN07757873 were excluded from further analysis due to their low expression of *Hepatocystis* genes (htseq-count reported below 50,000 reads mapped to the *Hepatocystis* assembly in each of these samples). The average filtered SNP counts per 10 kb of reference genome for each sample were calculated as the number of filtered SNPs divided by (genome size in kb * 10).

## RNA-seq deconvolution

Deconvolution of a bulk RNA-seq transcriptome sequence aims to determine the relative proportions of different cell types in the original sample. This requires a reference dataset of transcriptomes from "pure" cell types. To create this, we used single-cell *P. berghei* transcriptome sequences from the Malaria Cell Atlas [20]. For each cell type, single-cell transcriptome sequences were combined by summing read counts per gene to generate a set of pseudobulk transcriptome sequences (see our GitHub repository). The aim of summing across cells is to reduce the number of dropouts which are common in individual single-cell transcriptome

sequences. Bulk *Hepatocystis* RNA-seq transcriptome sequences, mapped and counted as above, were summed across replicates and filtered to exclude those with fewer than 100,000 reads. *Hepatocystis* and *P. berghei* pseudobulk read counts were converted to Counts Per Million (CPM) and *Hepatocystis* gene ids were converted to those of *P. berghei* one-to-one orthologues. Genes without one-to-one orthologues (defined by orthoMCL analysis) were excluded. CIBERSORT v1.06 [113] was used to deconvolute the *Hepatocystis* transcriptomes with the MCA pseudobulk as the signature matrix file. To test the accuracy of this deconvolution process we generated mixtures of the pseudobulk resulting in e.g. equal representation of read counts from male gametocyte, female gametocyte, ring, trophozoite and schizont pseudobulk transcriptomes (see our GitHub repository). We also deconvoluted bulk RNA-seq transcriptomes from Otto et al. [89] processed as in Reid et al. [19].

## Enrichment of missing genes in Malaria Cell Atlas gene clusters

We wanted to determine whether there were functional patterns common to orthologues missing from the *Hepatocystis* genome relative to *Plasmodium* species. To do this we looked for orthologous groups (orthoMCL as above) containing genes from *P. berghei*, *P. ovale wallikeri* and *P. vivax* P01, but not *Hepatocystis*. Genes from *P. berghei* have previously been assigned to 20 clusters based on their gene expression patterns across the whole life cycle [20]. We looked to see whether missing orthologues tended to fall into particular clusters more often than expected by chance (see our GitHub repository). We used Fisher's exact test with Benjamini-Hochberg correction to control the false discovery rate. We reported clusters with FDR $> = 0.05$.

## Supporting information

**S1 Fig. Conservation of synteny in the core regions of the assembly.** ACT (Artemis Comparison Tool) screenshot showing a comparison of centromere-proximal regions of *Hepatocystis* scaffold 132, *P. falciparum* 3D7 (Pf3D7) chromosome 4 and *P. vivax* (PvP01) chromosome 5. The red blocks represent sequence similarity (tBLASTx). The centromere is shown in green. Coloured boxes represent genes. The graph shows the GC-content.
(TIF)

**S2 Fig. Organization of putative subtelomeric regions of *Hepatocystis* scaffold 67, scaffold 211, *P. knowlesi* H chromosome 4 and *P. falciparum* 3D7 chromosome 9.** Exons are shown in coloured boxes with introns as linking lines. '//' represents a gap. The shaded/grey areas in *P. knowlesi* and *P. falciparum* mark the start of the conserved, syntenic regions to other *Plasmodium* species. The presence of genes that are subtelomeric in *Plasmodium* species, i.e. PHIST proteins, suggests that the *Hepatocystis* scaffolds are also subtelomeric. A complete subtelomere that includes telomeric repeats is missing in our *Hepatocystis* assembly. Thus, whether *Hepatocystis* chromosomes retain the organisation common to most *Plasmodium* species remains unclear.
(TIF)

**S3 Fig. Phylogenetic tree of *Haemosporidian* mitochondrial proteins.** *Hepatocystis* sp. ex. *Piliocolobus tephrosceles* (this work, marked with red arrow) appears next to a previously sequenced *Hepatocystis* sample from the flying fox *Pteropus hypomelanus* (NCBI accession FJ168565.1). Branches of the tree have been coloured by bootstrap support values from 45 (red) to 100 (green). Bootstrap values below 100 have also been added to the figure as text.
(TIF)

**S4 Fig. Phylogenetic tree of 18 apicoplast protein sequences of *Plasmodium* spp. and *Hepatocystis*.** Branches of the tree have been coloured by bootstrap support values from 66 (red) to 100 (green). Bootstrap values below 100 have also been added to the figure as text.
(TIF)

**S5 Fig. Phylogenetic tree of 11 nuclear genes of *Hepatocystis* and *Plasmodium* species.** Genes of *Hepatocystis* sp. ex *Piliocolobus tephrosceles* are highly similar to *Hepatocystis epomophori* genes sequenced in a different study [2]. The tree is based on the following genes: splicing factor 3B subunit 1, tubulin gamma chain, DNA polymerase delta catalytic subunit, eukaryotic translation initiation factor 2 gamma subunit, T-complex protein 1 subunit alpha, pantothenate transporter, ribonucleoside-diphosphate reductase large subunit, aminophospholipid-transporting P-ATPase, GCN20, transport protein Sec24A and RuvB-like helicase 3. Branches of the tree have been coloured by bootstrap values from 73 (red) to 100 (green). Bootstrap values below 100 have also been added to the figure as text. The red arrow points to the *Hepatocystis* sample from the current study.
(TIF)

**S6 Fig. Deconvolution using CIBERSORT and the Malaria Cell Atlas accurately determines the presence and absence of different *Plasmodium* life stages in bulk RNA-seq data.** (A) Pre-defined mixtures of pseudobulk RNA-seq data were deconvoluted with very high accuracy. (B) Real samples of *P. berghei* bulk RNA-seq from Otto et al (2014) were deconvoluted showing almost pure mixtures of gametocyte, ookinete or asexual stages as expected. The low proportions of expected parts of the IDC in each asexual sample may result from differences between what the MCA defines as a ring/trophozoite/schizont and what would microscopically be defined as such.
(TIF)

**S7 Fig. Multiple sequence alignments of two *Hepatocystis*-specific gene families.** (A) Alignment of *Hepatocystis*-specific gene family 1 (*hep1*). Pseudogenes (HEP_00099300, HEP_00250500, HEP_00323900) were not included in the alignment. HEP_00353700 is 476 amino acids long and was truncated here. (B) Alignment of *Hepatocystis*-specific gene family 2 (Hep2). This gene family contains a PEXEL motif (marked with a black box). Pseudogenes (HEP_00165000, HEP_00165200, HEP_00324000, HEP_00489100) were not included in the alignment.
(TIF)

**S8 Fig. Heatmaps of *Hepatocystis* gene family expression in the blood of its mammalian host.** (A) Expression levels (log vst-normalised) of *hep1* genes across blood samples from multiple red colobus monkeys. The estimated proportions of early blood stages (rings/trophozoites) and mature gametocytes are highlighted above. (B) Expression levels of *hep2* genes (C) Expression levels of *pir* genes.
(TIF)

**S9 Fig. Heatmap of *pir* protein subfamilies in *Hepatocystis* and *Plasmodium* species.** Rows correspond to species and columns correspond to *pir* subfamilies. The columns have been ordered by the number of sequences in each subfamily and the order of rows is approximately based on phylogeny. Colours represent the numbers of proteins belonging to each subfamily for each species. All *Hepatocystis pir* proteins belong to the only subfamily conserved across all these species [114] (indicated with red arrow).
(TIF)

**S10 Fig. Some orthologues missing in *Hepatocystis* sp. relative to *Plasmodium* species show common gene expression patterns across the *Plasmodium* life cycle.** (A) Malaria Cell Atlas (MCA) gene cluster 10 represents genes highly expressed in late schizonts. 25 genes from this cluster were conserved in *P. ovale wallikeri* and *P. vivax*, but were missing from our *Hepatocystis* genome assembly. Genes were clustered here by expression pattern and single-cells were ordered by pseudotime as in [20]. (B) MCA cluster 4 represents genes highly expressed across much of the life cycle—liver stages, trophozoites, female gametocytes and ookinetes/oocysts. 27 genes from this cluster were conserved in *P. ovale wallikeri* and *P. vivax*, but were missing from our *Hepatocystis* genome assembly.
(TIF)

**S11 Fig. Alignment of phylogenetic tree of putative RBP-related gene fragment in *Haemoproteus columbae*.** (A) Alignment of the translation of a sequence (GGWD01016989.1) from *Haemoproteus columbae* transcriptome assembly (GenBank: GGWD00000000.1) [23] with *Plasmodium* reticulocyte-binding proteins (RBPs) from PlasmoDB [115]. The alignment has been cropped to the length of the *Haemoproteus columbae* sequence. (B) Phylogenetic tree of *Plasmodium* RBPs and the *Haemoproteus columbae* sequence GGWD01016989.1, based on the alignment in panel A. Branch colours indicate bootstrap support values, from 33 (red) to 100 (green).
(TIF)

**S12 Fig. Distributions of *Hepatocystis* dN values in Malaria Cell Atlas (MCA) clusters.** *Hepatocystis* dN was calculated in 3-way comparison between *Hepatocystis*, *P. berghei* ANKA and *P. ovale curtisi* using codeml. The Malaria Cell Atlas clusters have been described in Fig 2B in the article on Malaria Cell Atlas [20]. (A) *Hepatocystis* genes with dN in the top 5%: observed versus expected ratios for Malaria Cell Atlas clusters. *Hepatocystis* genes that correspond to Malaria Cell Atlas clusters 2, 4 and 6 have less genes with dN rank in the top 5% than expected by chance (Fisher exact test p-value < 0.05). None of the MCA clusters contain significantly more genes ranked in the top 5% of dN than expected by chance, although there is a trend towards clusters 15 and 16 having higher dN. (B) Boxplot of all *Hepatocystis* dN values per each Malaria Cell Atlas cluster. Distribution of values in clusters 15 and 16 differs from the rest of the clusters. Kolmogorov-Smirnov test statistics are the following. Cluster 15 vs all other clusters: D = 0.42, p-value = 1.05e-05. Cluster 16 vs all other clusters: D = 0.52, p-value = 4.50e-12. Clusters 15 and 16 combined vs all other clusters: D = 0.46, p = 2.33e-15.
(TIF)

**S1 Table. Summary of gene properties.** For each gene in the assembly, the following is listed: annotation, number of exons, gene length (bp), the presence or absence of start and stop codons (reflecting the completeness of the assembly of the gene) and RNA-seq expression level (mean FPKM with standard deviation) in sample SAMN07757854 (RC106R). For the proteins encoded by the genes, the table shows the number of transmembrane segments predicted by TMHMM, ExportPred 2 score, 1 to 1 orthologs in *P. berghei* ANKA and *P. ovale curtisi* GH01 (based on OrthoMCL), PFAM domains, the number of matches to PEXEL motif (RxLxE/Q/D) and SignalP-5 signal peptide prediction.
(XLSX)

**S2 Table. Raw and normalised *Hepatocystis* gene expression data.**
(XLSX)

**S3 Table. *Plasmodium* orthologues missing in the *Hepatocystis* genome assembly.** *Plasmodium berghei* genes, which have an orthologue in *P. ovale curtisi* or *P. vivax*, but not the

*Hepatocystis* sp. DNA (A) or RNA-seq (B) assemblies and are enriched in Malaria Cell Atlas gene clusters.
(XLSX)

**S4 Table. Genes with *Hepatocystis* dN rank in the top 5% in codeml 3-way comparison between *Hepatocystis*, *P. berghei* ANKA and *P. ovale curtisi* GH01.** The total number of genes in dN analysis was 4009, out of which 200 correspond to 5%. The table includes Malaria Cell Atlas cluster numbers for each gene.
(XLSX)

# Acknowledgments

We would like to thank Alan Tracey for advice on genome assembly and J. Byaruhanga, P. Katurama, A. Mbabazi, A. Nyamwija, J. Rusoke, D. Hyeroba and G. Weny and the staff of Makerere University Biological Field Station for assistance in the field. Manoj Duraisingh provided helpful comments on the manuscript.

# Author Contributions

**Conceptualization:** Theo Sanderson, Adam J. Reid.

**Data curation:** Eerik Aunin, Ulrike Böhme.

**Formal analysis:** Eerik Aunin, Adam J. Reid.

**Investigation:** Eerik Aunin, Ulrike Böhme, Noah D. Simons, Adam J. Reid.

**Project administration:** Tony L. Goldberg, Nelson Ting, Colin A. Chapman.

**Resources:** Tony L. Goldberg, Nelson Ting, Colin A. Chapman.

**Software:** Eerik Aunin, Adam J. Reid.

**Supervision:** Nelson Ting, Matthew Berriman, Adam J. Reid.

**Visualization:** Eerik Aunin, Ulrike Böhme, Adam J. Reid.

**Writing – original draft:** Eerik Aunin, Ulrike Böhme, Theo Sanderson, Adam J. Reid.

**Writing – review & editing:** Eerik Aunin, Ulrike Böhme, Theo Sanderson, Noah D. Simons, Tony L. Goldberg, Nelson Ting, Colin A. Chapman, Chris I. Newbold, Matthew Berriman, Adam J. Reid.

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
