## [Decision Letter · Decision Letter 0]

12 Mar 2020

Dear Dr. Reid,

Thank you very much for submitting your manuscript "Genomic and transcriptomic evidence for descent from Plasmodium and loss of blood schizogony in Hepatocystis parasites from naturally infected red colobus monkeys" for consideration at PLOS Pathogens. As with all papers reviewed by the journal, your manuscript was reviewed by members of the editorial board and by several independent reviewers. In light of the reviews (below this email), we would like to invite the resubmission of a significantly-revised version that takes into account the reviewers' comments.

A brief editorial comment: I share reviewer 2's view that a complete genome assembly should not be a requirement for publication. The sparse parasite data gleaned from sequencing of the host has been effectively mined to answer the important questions posed. However, I agree with reviewer 1 that some comparison with the published Haemoproteus genome would strengthen the manuscript.

We cannot make any decision about publication until we have seen the revised manuscript and your response to the reviewers' comments. Your revised manuscript is also likely to be sent to reviewers for further evaluation.

Sincerely,

Tim J.C. Anderson

Guest Editor

PLOS Pathogens

Xin-zhuan Su

Section Editor

PLOS Pathogens

Kasturi Haldar

Editor-in-Chief

PLOS Pathogens

orcid.org/0000-0001-5065-158X

Michael Malim

Editor-in-Chief

PLOS Pathogens

orcid.org/0000-0002-7699-2064

Reviewer's Responses to Questions

**Part I - Summary**

Reviewer #1: The manuscript describes the first genomic and transcriptomic analysis of the haemosporidian parasite genus Hepatocystis. The authors generated a draft genome from sequence data that originally were designed to generate a reference genome of the primate host Piliocolobus tephrosceles and use transcriptomic data to investigate parasite life cycle characteristics in the blood stage in the vertebrate host.

Malaria parasites with the predominant species Plasmodium falciparum, are the causative agents of the severe tropical disease in humans. No vaccines with high efficacy could be developed against Plasmodium parasites to date. Plasmodium parasites feature a complex life cycle with distinct invasive and replicating stages and host switches and a single life cycle phase, the intra-erythrocytic replication of asexual blood stages, induces disease. This particular life cycle step is thought to be exclusive to species of the genus Plasmodium and represents the trait that is used to define the genus Plasmodium. Notably, all other haemosporidian parasite genera, even the closest relatives of Plasmodium, Hepatocystis parasites, lack the asexual multiplication in the blood. In the present manuscript the authors work to an understanding of the evolution of haemosporidian parasites and the evolution of erythrocytic merogony in particular by studying the genome of Hepatocystis in combination with the transcriptome of Hepatocystis in a comparative approach with the published Plasmodium datasets. As Hepatocystis presents the closest relative of mammalian Plasmodium species, the study of this genus is of significance for the entire research field.

In the introduction the authors introduce the genus Hepatocystis very comprehensively and highlight the differences to Plasmodium (in e.g. life cycle, hosts). The authors confirm the sister relationship of Hepatocystis with species of the Vinckeia Plasmodium clade that infect rodents, which has been proposed in previous studies. Without question one of the most important findings of the study is the fact that no transcriptomic evidence for schizonts in the blood has been found. However, the authors point out that the schizonts could sequester away from the bloodstream as seen in some Plasmodium species. The detection of novel gene families is another essential finding and will certainly be the subject of many subsequent studies. The low number of pir genes in Hepatocystis in comparison to species in the Vinckeia clade and the complete absence of the Reticulocyte Binding Protein family again underline the numerous differences relating to the differing life cycle despite the close phylogenetic relationship. With the conclusion about changes in specific genes that might relate to its adaptation to the Culicoides vector, the authors support previous hypotheses that vector shifts into different dipteran families are associated with significant genomic changes.

The methods used are described very detailed and comply with the best standards in this field. Several authors on this manuscript have been involved in different Plasmodium genome publications in the past and therefore belong to the experts in this field. This study/genome & transcriptome data provides yet another valuable resource for the entire malaria parasite research community. Therefore, I consider it the greatest weakness of the study that the authors did not aim for a publication of the full annotated genome and rather provide a draft genome. Without question, using the sequence data that had been generated to study the monkey genome, to assemble the Hepatocystis draft genome was a great opportunity and the interesting results of the study speak for themselves. But the effort to work towards a complete genome would have made the outcome even stronger and uncertainties in some results such as in the lack of expansion of specific gene families could be ruled out.

For the first time, a genomic study investigates a haemosporidian parasite species that infects mammals and does not belong to the genus Plasmodium. However, the authors do not compare their data to the published genome of Haemoproteus, another haemosporidian genus that also lacks the erythrocytic merogony and infects birds. In the last sentence of the manuscript, the authors point to the fact that genome sequences of Haemoproteus will provide more insights in the molecular biology and evolution of malaria parasites. Haemoproteus presents a basal taxon in the haemosporidian phylogeny and hence, an inclusion in the comparative genomic analysis is important and could reveal additional interesting findings. I recommend, if possible, to include the published Haemoproteus genome data in the comparative analysis.

Figures and Tables are comprehensive and especially the figures follow the design of previous Plasmodium publications (that have been authored/co-authored by some of the authors of the manuscript) and enable the reader to make comparisons.

Data availability/depositions has been carried out to a great extent and a previous version of this manuscript had been made available on bioRxiv.

Overall, the study/manuscript presents novel insights into the evolution of a neglected haemosporidian parasite genus that infects mammals and features a greatly modified life cycle in comparison to Plasmodium and therefore the results allow conclusions about the evolution of the entire group of malaria parasites. I consider the data and results of the study a valuable resource for the large Plasmodium parasite research community.

Reviewer #2: Aunin et al. present research on a parasite in the genus Hepatocystis based on data from a project conducted on its monkey host. The genome assembly from only "parasite contamination" of host data can not reach the standards of quasi-chromosome level genome reconstructions that have become custom in recent years. Additionally, the authors can't even determine the parasite species for which they assembled a genome. But - read on - Aunin et al. still present a fascinating manuscript, as these seeming shortcomings are irrelevant for a clearly defined research question.

Schizogony is the process of asexual replication of malaria parasites in the vertebrate host (e.g. human). During this process the host can suffer anaemia, as red blood cells are depleted. Aunin et al. seek an answer to the question whether Hepatocystis has lost blood schizogony. This has long been suggested, as asexual replicative forms (schizonts) have not been observed in parasites from this genus. If this question can be answered positively, the genetic correlates of presence/absence of such a process are interesting, as they could hint towards genetic pathways regulating blood schizogony in Plasmodium.

Genomic and transcriptomic analyses is presented based on samples from an undefined species. Does it matter? Not in my opinion! What matters is that we are presented with a parasite species sharing a more recent common ancestor with some Plasmodium species than those Plasmodium species share with each other. Aunin et al. resolve the phylogenetic placement of Hepatocystis as a sister to rodent Plasmodium species used as model systems in malaria research. But also P. ovale and P.vivax infecting humans are more closely related to Hepatocystis than to the most pathogenic human malaria parasite P. falciparum. In phylogenetic terms this means that Hepatocystis species are rendering the genus Plasmodium paraphyletic. Because of this phylogenetic context, we can learn about Plasmodium species from the single - even undetermined - Hepatocystis species analysed here.

A more complete and contiguous reconstruction of the genome would surely allow subtle questions on genome evolution. While this would certainly be beneficial for future research on other Hepatocystis species, I rather want to focus on whether the data is sufficient for a clear answer on the research question. This boils down to asking whether the genome is sufficiently well reconstructed to conclude that absence of certain gene families (i.e. those involved in blood schizogony) constitutes a "true negative" finding.

Relevant for this question are centromere-proximal and subtelomeric regions of the genome. Those contain not only replicated sequences notoriously hard to assemble, but also effector genes important for host-parasite interaction. The authors are open about the fact that telomeric repeat regions were not assembled and centromere-proximal regions were only assembled to a certain degree. Aunin et al., nevertheless, show that the fragmented genomic scaffolds recovered for those regions still contain an expected gene repertoire. They employ a "custom annotation" derived from single cell transcriptomics in a project called the malaria cell atlas (MCA) and demonstrate the absence (significant under-representation, more technically) of orthologues of "schizogony genes" in Hepatocystis sp..

This genomic data would not have been sufficient for strongly concluding the absence of blood schizogony. To strengthen their argument Aunin et al. were able to obtain Hepatocystis sp. transcriptomes from an impressive number (25 samples deemed usable based on coverage in an analysis of a total 29) of samples. RNASeq data had also originally been produced for the annotation of the monkey host's genome. Aunin et al. then used a clever analysis called "deconvolution", which characterises expression profiles by lifecycle stages. Essentially - again - in comparison to single cell (MCA) transcriptome profiles, this shows that the schizont expression profiles are absent in the Hepatocystis sp.'s transcriptome.

I am convinced the research question has been convincingly answered based on these two analyses. The analyses are performed expertly and potential problems are resolved with scrutiny. Aunin et al. present a number of interesting findings as genetic correlates with the loss of blood schizogony: a completely novel Hepatocystis specific gene family, expansion of gene families relevant for sporozoites (the lifecycle stage in the insect vector transmitting the parasite and in liver cells) and evolutionary divergence - likely by positive selection on non-synonymous sites - of genes involved in sexual replication and/or vector infection.

**Part II – Major Issues: Key Experiments Required for Acceptance**

Reviewer #1: As pointed out in the summary (Part I), a high-quality full genome of Hepatocystis would have been desirable, given the experience and expertise of several of the authors of this manuscript. Could the authors consider to invest additional work and aim for publication of a full genome? Could the authors provide reasons, why this is perhaps not feasible?

Again, as pointed out in the summary (Part I), why did the authors not compare their genomic Hepatocystis data to the published genome of the avian-infecting genus Haemoproteus? This basal haemosporidian taxon also lacks the asexual multiplication in the blood of their vertebrate hosts and a comparison to Hepatocystis could potentially reveal additional insights into the evolution of the entire malaria parasite group. Could the published Haemoproteus genome data be incorporated in the comparative analysis?

Reviewer #2: No major issues.

**Part III – Minor Issues: Editorial and Data Presentation Modifications**

Reviewer #1: Please find below some minor issues/suggestions

The authors state: “Species of the genus Hepatocystis are single-celled eukaryotic parasites infecting Old World monkeys, fruit bats and squirrels“

Species of Hepatocystis have also been described from the mammalian order Artiodactyla (e.g. Hippopotamus and Tragulus). So, I recommend adding “amongst others” or also listing this mammalian host group.

The authors state: “In contrast, liver merozoites of Hepatocystis spp. are thought to commit to the development of transmission stages directly upon invading red blood cells.“

Just for clarity, I recommend to specify that the transmission stages are sexual (gametocyte) stages.

The authors state: „They are then vectored not by mosquitoes, but by biting midges of the genus Culicoides“

To highlight that the Culicoides midges belong to a different family, the authors could add the family name (Ceratopogonidae).

One additional comment the vector: So far, Culicoides has only been confirmed as vector for the primate-infecting species Hepatocystis kochi. It still needs to be verified, if this or other Culicoides species transmit Hepatocystis in bats and the other mammalian groups.

The authors state: “At least four species of Hepatocystis are known to infect African monkeys – H. kochi, H. simiae, H. bouillezi and H. cercopitheci (6) – but with little sequence data currently linked to morphological identification, it was not possible to determine the species.”

Studies of the bat-infecting Hepatocystis species have proposed that bats in Africa and Australia are rather infected with parasites that belong to Hepatocystis species-complexes than different species. Could this also apply to the primate-infecting “species”? Perhaps the authors could comment on this.

The authors state: “Limited sequence data are available for Hepatocystis outside of this study, however 11 nuclear genes have been sequenced for H. epomophori, a parasite of bats (2). Based on the sequence of these genes, we found that HexPt forms a sister group to H. epomophori (S5 Fig; S1 Dataset).”

Limited sequence data for Hepatocystis is available and almost exclusively comprises short gene sequences for phylogenetic studies. However, it is probably still worth noting that some multiple-gene phylogenies investigated the phylogenetic placements within the Hepatocystis clade, showing that all Hepatocystis species from monkeys (from Asia and Africa) group in one monophyletic clade, sister to the Hepatocystis species that infect African bats. There is evidence that the Hepatocystis species of bats are not monophyletic and that Hepatocystis species from bat hosts of the genus Pteropus group basal to all other Hepatocystis species and therefore Hepatocystis species of monkeys might represent a derived clade.

The authors state: “In vivo transcriptome data supports a lack of erythrocytic schizogony. Transcriptome sequencing of blood samples from 29 individuals was performed as part of the red colobus monkey genome sequencing project (12). We found evidence that each of these individuals was infected with the same species of Hepatocystis as found in the genomic reads, consistent with high prevalence of this parasite in Kibale red colobus monkeys as previously reported (6)."

Mixed haplotype infections have been reported/investigated by Thurber et al. How did the authors deal with the mixed haplotype infections in their samples? Did the sequences that were detected within the monkey genome sequences show hints of a mixed infection?

The manuscript contains a few spelling errors/typos. In several cases, “spp.” is erroneously written italics (e.g. abstract, results) and “Vinckei” should be “Vinckeia”.

“Sample collection and data generation”

Have permits and agreements been expanded to allow use of DNA samples for parasite analysis in addition to the originally planned analysis of the monkey genome? (Nagoya/CBD)

Reviewer #2: The issues I see in the manuscript are relatively minor. I first want to highlight my perception of the special character of the manuscript as an asset: I consider the (very successful) re-use of host-genome "contamination" for parasite research a central aspect of the study. In my opinion this could be more clearly stated as not only viable but attractive for parasitologists working with genomic methods. The first sentence of the discussion could e.g. easily include a statement on data provenance. I also find the the first sentence of the methods could be improved (missing "a" before "project", instead of "a different project", rather "a project with the aim to generate" or similar). In my opinion the manuscript could be generally more pronounced on this point.

Taxonomy: The data is from only one species of the genus Hepatocystis. This is made clear in the context of the whole manuscript (esp. in the introduction "thus classified the parasite as Hepatocystis sp. ex Piliocolobus tephrosceles (HexPt; NCBI Taxonomy ID: 2600580)". In my opinion, nevertheless, Hepatocystis _sp._ should be used to refer to a - for the purpose of this manuscript undetermined - species in this genus throughout the manuscript (I'd prefer this over the more technical HexPt, but this is a matter of taste). As stated above, I am very positive about the general value of findings from this one species, as loss of blood schizogony is likely a major, sufficiently rare, evolutionary transition. Using _sp_ throughout the manuscript would still be more conservative in my opinion.

Apart from this, I think that the the figures could be improved to do full justice to findings:

Fig 1 A. This figure and the corresponding analysis could include additional information on the coverage profiles for the contigs/scaffolds with the different sequence similarity and GC (see e.g. "blobtools"). The signal of (low) GC content is expectedly strong for a Plasmodium sp./Hepatocystis sp. data set, it would be great to have sequencing coverage as a second visually distinguishing feature. This could also be (an) additional panel(s) for different sequencing libraries.

Fig 1 B. The presented colour legend should be changed/expanded to separate geographical information from host species information. Information on the latter should also be included for Plasmodium species, like in figure 2 (ideally using the same symbols).

Fig 2. In my opinion the phylogenetic tree - transporting the core message of the whole figure - should attract more attention in this. The "figure columns" "ap2 protein" and "tRNA" take (together) a similar amount of space as the tree but show relatively little variation, hence contain hardly any information.

- I am missing a figure for the paragraph "Missing orthologues tend to be involved in erythrocytic schizogony". This lacks representation in both main text figures or tables. Plotting expected/observed ratios for the number of orthologues found (like for SNPs in figure S11, but see comment below!) could potentially help to visualise missing or surplus orthologues in MCA clusters. Using this as an anchor (with Hepatocystis sp. data), visualisation of "Cluster 4" and "Cluster 10" from MCA relying on P.berghei data (Figure S10) could be included as additional panels. This might help to introduce the interpretation of MCA clusters for the present manuscript generally. MCA clusters play an important role as "custom annotation" in two parts of the analysis: enrichment of genes missing in Hepatocystis sp. and enrichment of high-dN gens in those clusters. It could be nice to introduce MCA more visually this way as part of a main manuscript figure!

Fig 3. The caption of this figure only refers to figure panel B (and somewhat to C), but not to A. For A, I am also missing the information (in the legend) that the SNPs are called relative to the genome assembly(?). I am wondering whether A could be a figure on its own (SNPs in the transcriptome), maybe with an additional panel presenting the enrichment of SNPs in the MCA clusters (currently S11)? Btw. Fig S11 A should make more clear that a ratio of 1 is the null expectation: rotation by 90° and plotting deviations from 1 (as bars or lines) would help to make this more obvious. Then current 3 B and C could be separate figure committed to (deconvoluted) expression patterns in the transcriptome and all panles would fit the caption.

Table 2 should, in my opinion, contain context for the expected size of gene families. Listing the family size from Plasmodium species (e.g. P. berghei, P. vivax and P. falciparum; just like in the genome comparisons of Table 1) would add such context. Also: "Frequency of members" sounds a bit too technical to me, isn't this simply the "gene family size"?

PLOS authors have the option to publish the peer review history of their article (what does this mean?). If published, this will include your full peer review and any attached files.

Reviewer #1: No

Reviewer #2: Yes: Emanuel Heitlinger
---

## [Decision Letter · Decision Letter 1]

19 Jun 2020

Dear Dr. Reid,

We are pleased to inform you that your manuscript 'Genomic and transcriptomic evidence for descent from Plasmodium and loss of blood schizogony in Hepatocystis parasites from naturally infected red colobus monkeys' has been provisionally accepted for publication in PLOS Pathogens.

Before your manuscript can be formally accepted you will need to complete some formatting changes, which you will receive in a follow up email.  A member of our team will be in touch with a set of requests. In addition, please address the minor edits requested by reviewer #2.

Best regards,

Tim J.C. Anderson

Guest Editor

PLOS Pathogens

Xin-zhuan Su

Section Editor

PLOS Pathogens

Kasturi Haldar

Editor-in-Chief

PLOS Pathogens

orcid.org/0000-0001-5065-158X

Michael Malim

Editor-in-Chief

PLOS Pathogens

orcid.org/0000-0002-7699-2064

Thanks for the care taken in thoroughly addressing the reviewers comments. I have no further comments and think this is an excellent paper

Reviewer Comments (if any, and for reference):

Reviewer's Responses to Questions

**Part I - Summary**

Reviewer #1: The manuscript is substantially improved and clarified in the revised version and the authors should be commended for responding so positively to the reviewer's comments. All the major issues have been clarified.

Reviewer #2: As mentioned in my first review I find that Aunin et al. present an outstanding manuscript and I recommend it for publication in its present form with very minor revisions, if any at all.

Reports finding that databases are ripe contamination wrongly assigned to target species are ample. Such contamination can be problematic for the assignment of taxonomic origin in metagenomics or when gene family evolution is reconstructed [1,2]. When not scrutinised in genome projects even simple contamination from co-cultured organisms can lead to wrong conclusions, such as claims about astonishingly prevalent horizontal gene transfer into eukaryote genomes [3]. A study on contamination of animal genomes found that high amounts of sequences derived from parasites and symbiont are present in genome assemblies. When treated correctly sequence data from such pathogens and symbionts, in contrast to culture contamination, presents a biological signal worth further investigation. Remarkably the parasite contamination in animal genomes was found be be derived mainly from Apicomplexan parasites, providing opportunities especially for genomics in parasitology [4].

Aunin et al. take such genomic data re-use in parasitology to a novel and extraordinary level: they present research on a single parasite species reconstructing a relatively complete genome and transcriptome. Aunin et al. present the genome of Hepatocystis sp., a relative of Plasmodium species. They can - based on the gonome this one species - confirm previous findings, that Hepatocystis is rendering the genus Plasmodium paraphyletic. This means that a common ancestor of species of the two genera is found more recently than common ancestors with e.g. P. falciparum, the causative agent of Malaria tropica. In contrast to the other species in this common clade, species in the genus Hepatocystis are believed to have lost asexual stages (schizonts) replicating in red blood cells.

From mere contamination of the red colobus monkey's genome Aunin et al. construct a parasite genome sufficiently complete to conclude on even the absence of gene families as a "true negative" finding: Reticulocyte binding proteins (RBPs) a gene family involved in schizogony (the process producing erythrozytic schizonts) is found absent from the Hepatocystis sp. genome.

Aunin et al. employ a "custom annotation" derived from single cell transcriptomics. Briefly, data from a project called the malaria cell atlas (MCA), is used to classify Plasmodium genes regarding their expression in particular lifecycles stages. Using this classification Aunin et al demonstrate the absence (significant under-representation, more technically) of orthologues of "schizogony genes" genes in genome and transcriptome assemblies of Hepatocystis sp.. Taken together the absence of "blood schizogony genes" in relatively complete genome assemblies, and the lack of expression of those genes in blood, the tissue those genes are expected to be expressed most in, provides sufficient evidence for absence. This delivers genomic correlates with the absences of blood schizogony. As this process is causing most virulence of Plasmodium species in their vertebrate hosts, this is an important finding.

[1] Florian P. Breitwieser, Mihaela Pertea, Aleksey V. Zimin, and Steven L. Salzberg. Human contamination in bacterial genomes has created thousands of spurious proteins. Genome Res. June 2019 29: 954-960; Published in Advance May 7, 2019, doi:10.1101/gr.245373.118

[2] Clementine M. Francois, Faustine Durand, Emeric Figuet and Nicolas Galtier. Prevalence and Implications of Contamination in Public Genomic Resources: A Case Study of 43 Reference Arthropod Assemblies. G3: Genes, Genomes, Genetics February 1, 2020 vol. 10 no. 2 721-730; https://doi.org/10.1534/g3.119.400758

[3] Georgios Koutsovoulos, Sujai Kumar, Dominik R. Laetsch, Lewis Stevens, Jennifer Daub, Claire Conlon, Habib Maroon, Fran Thomas, Aziz A. Aboobaker, Mark Blaxter No evidence for extensive horizontal gene transfer in the genome of the tardigrade Hypsibius dujardini. Proceedings of the National Academy of Sciences May 2016, 113 (18) 5053-5058; DOI: 10.1073/pnas.1600338113

[4] Borner, J., Burmester, T. Parasite infection of public databases: a data mining approach to identify apicomplexan contaminations in animal genome and transcriptome assemblies. BMC Genomics 18, 100 (2017). https://doi.org/10.1186/s12864-017-3504-1

**Part II – Major Issues: Key Experiments Required for Acceptance**

Reviewer #1: (No Response)

Reviewer #2: None.

**Part III – Minor Issues: Editorial and Data Presentation Modifications**

Reviewer #1: (No Response)

Reviewer #2: Individual points in the revision after my first review (all my points I don't comment on have been fully addressed, I also comment on one more than fully addressed point) :

- "We had some doubts about the use of HexPt and have altered this throughout the manuscript to the less intrusive Hepatocystis sp." I find this a good decission. However, I still find a few occasions of "Hepatocystis" (without the "sp.") referring clearly to only the one species studied. E.g. "In fact, we could not identify any RBPs in the Hepatocystis genome or Hepatocystis RNA-seq assemblies." It's probably pedantic, but I would appreciate to clearly refer to the one species studied in those and similar cases using the "sp.". There are other occurrences were Hepatocystis clearly is meant as a genus, and a "twilight zone" where either the one species is meant or an extrapolation to mean the whole genus is made. Clearly the decision where this extrapolation is suitable can be left to the authors.

- I find that the 2D density plots integrating coverage profiles with density distributions of GC content add information. I assume the high coverage - low GC contigs are from the apicoplast genome of Hepatocystis sp.. I fully understand that the authors want to use the figure in it's present form to indicate what they used in this study as sufficient to sort out Hepatocystis contigs. I only remark that the visualisation of the very prominent high coverage apicoplast contigs could help guide others to detect apicomplexan "genome contaminants". This might be relevant in apicomplexans with higher GC content.

- "We have redrawn Figure 2B from the MCA paper, using the supplementary data and added the observed/expected ratios used in calculating the clusters with unexpectedly high levels of missingness in Hepatocystis. This has been added as a new Figure 3 in the main manuscript." It's rather Figure 4 not 3. I find this to be a very nice and informative figure now. I much appreciate how the enrichment scores (great to use log ratios!) are aligned with the heatmap columns! Much better than what I imagined when I recommended a figure in this style. Absolutely great figure

PLOS authors have the option to publish the peer review history of their article (what does this mean?). If published, this will include your full peer review and any attached files.

Reviewer #1: Yes: Juliane Schaer

Reviewer #2: Yes: Emanuel Heitlinger

---

## [Editor Report · Acceptance letter]

15 Jul 2020

Dear Dr. Reid,

We are delighted to inform you that your manuscript, "Genomic and transcriptomic evidence for descent from Plasmodium and loss of blood schizogony in Hepatocystis parasites from naturally infected red colobus monkeys," has been formally accepted for publication in PLOS Pathogens.

Best regards,

Kasturi Haldar

Editor-in-Chief

PLOS Pathogens

orcid.org/0000-0001-5065-158X

Michael Malim

Editor-in-Chief

PLOS Pathogens

orcid.org/0000-0002-7699-2064